# Sanity Checks for Lottery Tickets: Does Your Winning Ticket Really Win the Jackpot?

**Xiaolong Ma**[1,†], **Geng Yuan**[1,†], **Xuan Shen**[1], **Tianlong Chen**[2], **Xuxi Chen**[2], **Xiaohan Chen**[2], **Ning Liu**[3], **Minghai Qin**, **Sijia Liu**[4], **Zhangyang Wang**[2], **Yanzhi Wang**[1]

[1] Northeastern University, [2] University of Texas at Austin
[3] Midea Group, [4] Michigan State University
{ma.xiaol,yanz.wang}@northeastern.edu

## Abstract

There have been long-standing controversies and inconsistencies over the experiment setup and criteria for identifying the "winning ticket" in literature. To reconcile such, we revisit the definition of lottery ticket hypothesis, with comprehensive and more rigorous conditions. Under our new definition, we show concrete evidence to clarify whether the winning ticket exists across the major DNN architectures and/or applications. Through extensive experiments, we perform quantitative analysis on the correlations between winning tickets and various experimental factors, and empirically study the patterns of our observations. We find that the key training hyperparameters, such as learning rate and training epochs, as well as the architecture characteristics such as capacities and residual connections, are all highly correlated with whether and when the winning tickets can be identified. Based on our analysis, we summarize a guideline for parameter settings in regards of specific architecture characteristics, which we hope to catalyze the research progress on the topic of lottery ticket hypothesis. Our codes are publicly available at: https://github.com/boone891214/sanity-check-LTH.

## 1 Introduction

In recent years, the Lottery Ticket Hypothesis (LTH) [1] has drawn great attention and thorough research efforts. As an important study to investigate the initialization state and network topology of the deep neural networks (DNNs), LTH claims the existence of a winning ticket (i.e., a properly pruned subnetwork together with original weight initialization) that can achieve competitive performance to the original dense network, which highlights great potential for efficient training and network design.

Unfortunately, among the various researches on the lottery ticket hypothesis [2, 3, 4, 5, 6, 7, 8], there are many inconsistencies regarding the settings of training recipe, and they further lead to the controversies over the conditions for identifying winning tickets. We revisit and analyze the definition of the original lottery ticket hypothesis and find that the quality of training recipe is a critical factor for the network performance, which in fact, is largely missing in previous discussions.

In the standard LTH setup [1], key training hyperparameters such as learning rate and training epochs were not scrutinized nor exhaustively tuned. The winning ticket can be identified in the case of small learning rate, but can fail to emerge at higher initial learning rates especially in deeper networks. For instance, in [1], the winning tickets can be identified only in the case of small learning rate 0.01, with ResNet-20 and VGG-19 on CIFAR-10. At larger learning rates, however, [9] reveals that the "winning ticket" has no accuracy advantage over the random reinitialization , which contradicts with

---

† Equal contribution.

35th Conference on Neural Information Processing Systems (NeurIPS 2021).

the LTH definition. On the other hand, the settings in [1] train 78 epochs for ResNet-20 on CIFAR-10. Such insufficient training causes a relatively low pretraining accuracy. When pruned iteratively, the subnetwork accuracy can easily match that pretraining accuracy of the original network. Under such experimental conditions, the existence of the winning ticket is questionable.

In addition to all the problems caused by the experimental conditions, the huge computational consumption to find a winning ticket becomes another research barrier and the practical main drawback, limiting the observations made on LTH. For instance, to reach around 90% overall sparsity ratio, iterative magnitude-based pruning (IMP) in [1] requires totally 11 iterations (20% of the weights are pruned in each iteration). It adds up to 1,760 total training epochs if each iteration consumes 160 epochs. On the other hand, as an efficient pruning method, one-shot magnitude-based pruning (OMP) prunes a pretrained DNN model to arbitrary target sparsity ratio in one shot, which greatly saves training efforts. However, OMP is rarely considered in the related literature, and is often deemed as "weak" without full justification. Based on the above reasons, we feel we cannot confidently draw arguments, before we are able to evaluate LTH comprehensively in regards of key factors such as different network structures, network dimensions, and training dataset sizes.

In this paper, we dive deeper into the underlying condition of the lottery ticket hypothesis. We raise the following questions: (1) What makes the comprehensive condition to define the lottery ticket hypothesis? (2) Do winning tickets exist across the major DNN architectures and/or applications under such definition? and (3) What are the intrinsic reasons for their existence or non-existence?

To answer the above questions, we present our rigorous definition of the lottery ticket hypothesis, which specifies settings of the training recipe, the principles for identifying winning tickets, and the rationality on examining the winning ticket existence. Under this rigorous definition, we perform extensive experiments with many representative DNN models and datasets. The relationships between winning tickets and various factors are quantitatively analyzed. We empirically study the patterns through our analysis, and develop a guideline to ease the process of obtaining the winning ticket. Our findings open up many new questions for future work. We summarize our contributions as follows:

**I.** We point out that the usage of inappropriately small learning rates, insufficient training epochs, and other inconsistent and implicit conditions for identifying winning ticket in the literature, are the main reasons that cause controversies in the lottery ticket studies.

**II.** We propose a more rigorous definition of the winning ticket, and evaluate the proposed definition on different training recipe, DNN architecture, network dimension, and the training data size. Somehow surprisingly, we find that under the new rigorous definition, *no* "rigorous" winning tickets are found by current methods, while there do exist winning tickets under a slightly looser definition.

**III.** We find that when residual connections exist in the network, using a relatively small learning rate is more likely to find (close to) winning tickets. When no residual connection exists, the IMP method may not be necessary because OMP can achieve equivalent performance.

**IV.** We also find that when a smaller learning rate is not favorable, initialization is likely to make no difference in finding the winning ticket (e.g., lottery initialization is not necessary). We quantitatively analyze the patterns, and present a guideline to help identify winning tickets.

## 2 Re-defining Lottery Ticket Hypothesis

### 2.1 Notations and Preliminary

In this paper, we follow the notations from [1, 5]. Detailed notations and functions are listed in Table 1. Based on Table 1, we provide several key LTH-related settings along with descriptions.

Consider a network function $f(\cdot)$ that is initialized as $f(x; \theta_0)$ where $x$ denotes input training samples. We define the following settings:

- *Pretraining:* We train the network $f(x; \theta_0)$ for $T$ epochs, arriving at weights $\theta_T$ and network function $f(x; \theta_T)$.

- *Pruning*: Based on the trained weights $\theta_T$, we adopt $\text{OMP}(\theta_T, s)$ or $\text{IMP}(\theta_T, s)$ to generate a pruning mask $m_O, m_I \in \{0, 1\}^{|\theta|}$. Note that for IMP, the same $\theta_0$ is used in each iteration to ensure fairness to OMP.

Table 1: Summary of notations and functions.

| Notation | Description |
|---|---|
| $T$ | $T$ is the total number of training epochs. |
| $\theta_0, \theta_t, \theta_0'$ | $\theta_0 \sim \mathcal{D}_\theta$ denotes initial weights used for training. $\theta_t$ is the weights that is trained from $\theta_0$ for $t$ epochs where $t \leq T$. $\theta_0' \sim D_\theta$ denotes a random reinitialization that is different from $\theta_0$. |
| $m$ | A sparse mask $m \in \{0, 1\}^{|\theta|}$ is obtained from certain pruning algorithm. |
| $s$ | $s$ is the sparsity ratio, which is defined as the percentage of pruned weights in the DNN model. |
| $\theta^{SD}$ | $\theta^{SD}$ denotes the weight in a small-dense model that has the same number of non-zero parameters as a pruned model, i.e. $\theta^{SD} \sim \mathcal{D}_{||m||}$. |
| $\mathrm{OMP}(\theta, s)$ | One-shot Magnitude-based Pruning [10] that prunes $\theta_T$ and returns $m$, i.e. $m_O \leftarrow \mathrm{OMP}(\theta_T, s)$. It prunes $s \times 100\%$ of weights in a one-time operation manner. |
| $\mathrm{IMP}(\theta, s)$ | Iterative Magnitude-based Pruning [11] that prunes $\theta_T$ and returns $m$, i.e. $m_I \leftarrow \mathrm{IMP}(\theta_T, s)$. $\mathrm{IMP}(\cdot)$ prunes 20% of remaining weights per iteration until arriving at target sparsity $s$ [5]. |

- *Lottery ticket with $\mathrm{OMP}$ (LT-$\mathrm{OMP}$):* We directly apply mask $m_O$ to initial weights $\theta_0$, resulting in weights $\theta_0 \odot m_O$ and network function $f(x; \theta_0 \odot m_O)$.

- *Lottery ticket with $\mathrm{IMP}$ (LT-$\mathrm{IMP}$):* We apply $m_I$ to initial weights $\theta_0$, and get $f(x; \theta_0 \odot m_I)$.

- *Random reinitialization with $\mathrm{OMP}$ (RR-$\mathrm{OMP}$):* We apply mask $m_O$ to the random reinitialized weights $\theta_0'$, and get network function $f(x; \theta_0' \odot m_O)$.

- *Random reinitialization with $\mathrm{IMP}$ (RR-$\mathrm{IMP}$):* We apply $m_I$ to random reinitialized weights $\theta_0'$, and get $f(x; \theta_0' \odot m_I)$.

- *Small-dense training (SDT):* We construct a small-dense network that has the same depth and reduced width compared to the original network, and initialized by $\theta^{SD}$, i.e. $f(x; \theta^{SD})$.

**Original definition of the winning ticket**: The original lottery ticket hypothesis [1] claims that there exists subnetwork $f(x; \theta_0 \odot m)$ in a randomly initialized dense network $f(x; \theta_0)$, that once trained for $T$ epochs (or fewer) will result in similar accuracy as $f(x; \theta_T)$, under a non-trivial sparsity ratio. Additionally, the accuracy of $f(x; (\theta_0 \odot m)_T)$ should be noticeably higher than $f(x; (\theta_0' \odot m)_T)$. Note that $(\theta_0 \odot m)_T$ and $(\theta_0' \odot m)_T$ are the initial and the randomly reinitialized weights of the sparse subnetwork trained for $T$ epochs, respectively. When the above conditions are met, $(\theta_0 \odot m)$ can be considered the *Winning Ticket*.

We define a network is *well-trained*, if it is trained using a sufficient training recipe (i.e., an appropriate learning rate and sufficient training epochs). However, in many prior works such as [1], the pretraining of the lottery ticket experiments used an insufficient training recipe (i.e., inappropriately small learning rate and fewer training epochs), which leads to non-optimal pretraining accuracy at relatively low levels. Apparently, a higher pretraining accuracy is more difficult for a subnetwork to match or "win the ticket", even by using a sufficient training recipe.

We further revisit the LT-$\mathrm{IMP}$ and RR-$\mathrm{IMP}$ experiments using ResNet-20 on CIFAR-10 dataset, at three different learning rates over a range of different sparsity ratios ([1] uses the small learning rate 0.01). We train the subnetworks with the same training recipe in pretraining, and we also adopt the settings in [1] to reproduce the results. Our preliminary results are shown in Figure 1.

Through Figure 1(a), 1(b), our first observation is that, under either training recipe, the "winning ticket" exists in smaller learning rates (e.g., 0.005 and 0.01), but does not exist at a relatively larger learning rate (e.g., 0.1). For instance, in the cases of the initial learning rate of 0.005 and 0.01, we find a noticeable accuracy gap between LT-$\mathrm{IMP}$ and RR-$\mathrm{IMP}$ using both training recipes, and the LT-$\mathrm{IMP}$ accuracy is close to the pretraining accuracy with a reasonable sparsity ratio (e.g., 50% or above). This is similar to the observations found in [1] on the same network and dataset. On the other hand, in the case of the initial learning rate of 0.1, the LT-$\mathrm{IMP}$ has a similar accuracy performance as the RR-$\mathrm{IMP}$, and cannot achieve the accuracy close to the pretrained DNN with a reasonable sparsity ratio, thus no winning ticket condition is satisfied.

Through Figure 1(c), our second observation is that, at the same learning rate, the winning ticket defined in [1] can be identified by using an insufficient training recipe, but fails to satisfy the winning

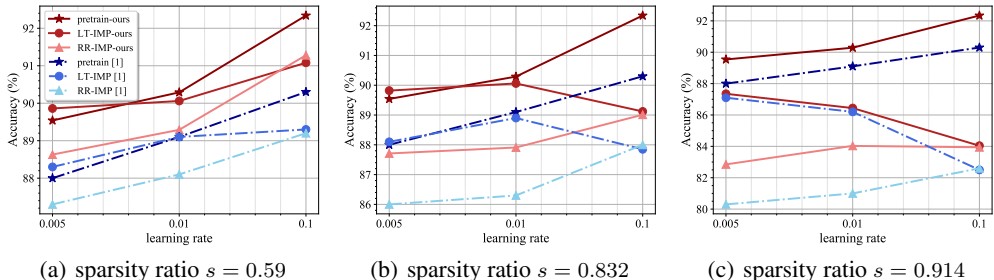

| (a) sparsity ratio $s = 0.59$ | (b) sparsity ratio $s = 0.832$ | (c) sparsity ratio $s = 0.914$ |

Figure 1: Preliminary results of ResNet-20 on CIFAR-10 dataset with different learning rates and sparsity ratios. We train the network using 160 epochs, while [1] uses 78 epochs. Please refer to [1] and Appendix A for the full results of all sparsity levels.

ticket condition when the network is well-trained. For instance, in the case of initial learning rate of 0.005, [1] uses approximately 78 epochs for training the network, which achieves 88.0% pretraining accuracy, 87.1% on LT-`IMP` and 80.3% on RR-`IMP`, respectively. The LT-`IMP` accuracy is close to the pretraining accuracy, and outperforms RR-`IMP`, thus it is claimed in [1] that the winning ticket is found. However, when we train the network with a sufficient number of epochs (160 in our settings), the accuracy of pretraining, LT-`IMP`, and RR-`IMP` are 89.6%, 87.4%, and 82.9%, respectively. In this case, the accuracy gap between pretraining and LT-`IMP` is not small enough to claim that they are "similar", thus in fact no winning ticket is found.

**Takeaway:** The above two observations indicate that the winning tickets are more likely to exist at a small learning rate or at an insufficient training epochs, but may not exist at a relatively large learning rate or sufficient training epochs (also observed in [9]). However, we would like to point out that using a relatively large learning rate (e.g., 0.1) and sufficient training epochs (e.g., 160, which is the standard settings on CIFAR-10) result in a *notably higher accuracy* for the pretrained DNN (92.3% vs. 88.0%). This point is largely missing in the previous discussions, and questions whether the previously identified "winning tickets" are meaningful enough.

## 2.2 A Rigorous Definition of the Lottery Ticket Hypothesis

The above discussion reveals the inconsistency of identifying the winning ticket under different conditions. We provide a more rigorous definition of lottery ticket hypothesis to reconcile the long-standing winning ticket identification discrepancy between experiment settings[1]. Our goal is to investigate the precise conditions on when winning ticket exists and how to identify them.

> **The lottery ticket hypothesis – a rigorous definition.** *Under a non-trivial sparsity ratio, there exists an identically initialized subnetwork that – when trained in isolation with a decent learning rate – can reach similar accuracy with the well-trained original network using the same or fewer iterations, while showing clear advantage in accuracy compared to a randomly reinitialized subnetwork as well as an equivalently parameterized small-dense network.*

**The principles for the identification of the winning tickets.** From our preliminary results in Figure 1, we recognize that the pretraining of the randomly initialized dense network $f(x; \theta_0)$ with different initial learning rates achieves varying accuracy. Based on this observation and the rigorous definition of lottery ticket hypothesis, we list the conditions for identifying winning ticket as follows:

① A non-trivial sparsity ratio $s$ and a sufficient training epochs $T$ are adopted for the subnetwork.

② SDT of $f(x; \theta_T^{SD})$ shows clear accuracy drop compared to the well-trained subnetwork.

③ There exists a learning rate such that the subnetwork $f(x; (\theta_0 \odot m)_T)$ achieves notably higher accuracy (with a clear gap) than $f(x; (\theta_0' \odot m)_T)$ trained with any learning rates.

④ There exists a learning rate such that the subnetwork $f(x; (\theta_0 \odot m)_T)$ achieves accuracy similar to or higher than the pretrained network $f(x; \theta_T)$ at the same learning rate.

---

[1]We also provide a mathematical construct in Appendix C.

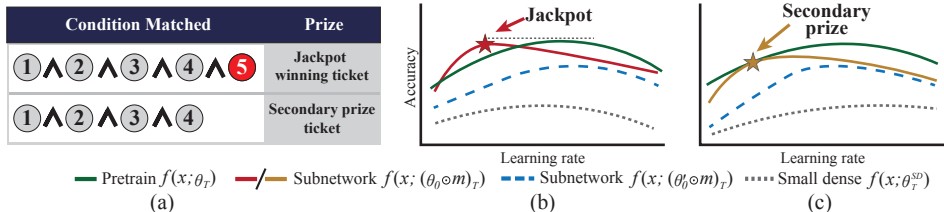

Figure 2: An illustration of the principles for identification of the winning tickets.

⑤ There exists a learning rate such that the subnetwork $f(x; (\theta_0 \odot m)_T)$ achieves accuracy similar to or higher than the *well-trained* original network $f(x; \theta_T)$ (i.e., trained with an appropriate learning rate and sufficient number of training epochs).

Our listed conditions *complete* the long missing but necessary aspects for identifying the winning ticket. ① formally recognizes the practical significance of the winning tickets, that a found network topology of the winning ticket should benefit the training/inference speed. It is commonly acknowledged that the overall sparsity ratio of the non-structured sparsity should exceed approximately 60% to deliver on-device acceleration. ② avoids a situation where the accuracy of the winning ticket is comparable to that of a small-dense network due to the over-parameterization of a network, which ensures the necessity of the winning ticket existence. ③ takes into account of the influences by different learning rates, which is missing in previous discussions. ④ is the original condition for identifying winning ticket in previous works, but it does not consider the best pretraining accuracy at a desirable learning rate. ⑤ takes the desirable training recipe into consideration, which is different from existing works and becomes the most crucial condition in our definition. We define "similar accuracy" as within 0.5% accuracy drop for CIFAR-10, 1% for CIFAR-100 and Tiny-ImageNet, and 1.5% for ImageNet-1K, and a "clear gap" between $f(x; (\theta_0 \odot m)_T)$ and $f(x; (\theta'_0 \odot m)_T)$ (condition ③) should be an accuracy difference over 0.5%.[2]

We summarize the principles for identifying the winning tickets in Figure 2 (a).

- In the case that a subnetwork $f(x; (\theta_0 \odot m)_T)$ satisfies the condition ① - ⑤ as Figure 2 (b) shows, we call $(\theta_0 \odot m)$ as *Jackpot winning ticket*, for it has the potential to completely match the best performance of the original dense network.

- On the other hand, the original "winning ticket" discussed in [1] achieves the pretraining accuracy that is clearly lower than the best pretraining accuracy as Figure 2 (c). In this case, condition ① - ④ are satisfied while the condition ⑤ is not, and we consider it as a *secondary prize ticket*.

We distinguish our definition of the lottery ticket hypothesis from the weight rewinding technique [5, 12]. Lottery ticket hypothesis, on one hand, is a study of initialization state and network topology for a neural network, while weight rewinding, on the other hand, studies the trade-off between accuracy and subnetwork searching cost. Despite the difference, we can generalize the weight rewinding technique into the winning ticket identification principle, which is shown in Appendix B. Detailed experimental evaluations of weight rewinding can also be found in Appendix D.

## 3 Sanity Checks for Lottery Tickets: Evaluation, Analysis and Guideline

Based on the rigorous definition of the lottery ticket hypothesis, we evaluate the lottery tickets with different types of network architectures, datasets with different sizes, and different learning rates. Detailed analysis are demonstrated for a deeper understanding of the lottery ticket hypothesis.

### 3.1 A Comprehensive Study Under the Rigorous Definition

**Networks and datasets:** In this section, we evaluate the lottery ticket hypothesis with various combinations of networks and datasets. We choose different network architectures among ResNet series [14], VGG [15], and MobileNet-v1 [16]. Specifically, the ResNet-32 is a wide version [17] with a width multiplier of 2. CIFAR-10/100 [18], Tiny-ImageNet [19] and ImageNet-1K [20] are all evaluated. Table 2 lists the details of the networks and datasets in the experiments we perform.

---

[2]Our quantitative criteria for accuracy gaps are no different from many previous efforts [3, 8, 7, 12, 13].

Table 2: Dataset and network we evaluate using the re-definition of the lottery ticket hypothesis.

| Dataset | CIFAR-10 | | CIFAR-100 | | | Tiny-ImageNet | | ImageNet-1K | |
|---|---|---|---|---|---|---|---|---|---|
| #Images | 50K/10K | | 50K/10K | | | 100K/10K | | 1.28M/50K | |
| #Classes | 10 | | 100 | | | 200 | | 1000 | |
| Img Size | $32 \times 32$ | | $32 \times 32$ | | | $64 \times 64$ | | $224 \times 224$ | |
| Network | RN-20 | RN-32 | MBNet-v1 | RN-18 | VGG-16 | RN-18 | RN-50 | RN-18 | RN-50 |
| #Params. | 0.27M | 1.86M | 3.21M | 11.22M | 14.72M | 11.68M | 25.56M | 11.69M | 25.56M |

Table 3: Summary of the observations of all experiments.

| Dataset | RN20 | RN32 | MBNet-v1 | RN18 | VGG-16 | Dataset | RN18 | RN50 |
|---|---|---|---|---|---|---|---|---|
| CIFAR-10 | ✗ ✓ ✓ ✓ | ✗ ✓ ✓ ✓ | △ ✓ ✗ ✗ | ✗ ✓ ✓ ✗ | ✗ ✗ ✗ ✗ | Tiny-ImageNet | ✗ ✗ ✗ ✗ | ✗ ✗ ✗ ✗ |
| CIFAR-100 | ✗ ✓ ✓ ✓ | ✗ ✓ ✓ ✓ | ✗ ✗ ✗ ✗ | △ ✓ ✓ ✓ | ✗ ✗ ✗ ✗ | ImageNet-1K | ✗ ✗ ✗ ✗ | ✗ ✗ ✗ ✗ |

■ Jackpot  ■ Secondary  ■ Prefer small lr  ■ Prefer IMP    ✓ Yes  ✗ No  △ At boundary

**Experimental setups:** In this paper, we conduct our experiments using different learning rates. We empirically set the (initial) learning rate from extremely small to normal, then to very large based on the network and dataset. At each learning rate, we conduct a series of experiments described in Section 2.1, and each experiment is run *three* times. For IMP(·), we follow the settings in [1, 5] that 20% of the weights are pruned in each iteration. For OMP(·), we directly prune the network to the same sparsity ratio as IMP(·). On CIFAR-10/100, We train the network for 160 epochs and the learning rates decrease by a factor of 10 after 80 and 120 epochs. On ImageNet-1K, We train the network for 90 epochs and cosine annealing learning rate schedule is used. We conduct our experiments on NVIDIA A100 with 8 GPUs. Detailed experiment settings are listed in Appendix E.

We plot the accuracy vs. learning rate curves for all experiments we run, and demonstrate them in Figure 3. Due to the space limits, we put the full results for all other networks, datasets and sparsity ratios in Appendix F.1. Based on the results, we summarize the observations in Table 3 and answer the following questions with detailed analysis. For the following discussion, if not otherwise specified, we use LT to denote the setting of the subnetwork training with LT-IMP or LT-OMP, and RR for RR-IMP or RR-OMP.

**Do Jackpot winning tickets exist in our evaluation?**

We carefully examine all the results. Unfortunately, under the rigorous definition of the lottery ticket hypothesis and current ticket searching methods (IMP(·) and OMP(·)), no clear Jackpot winning tickets are found, and even tickets that merely reach the boundary of conditions rarely exist. According to the experiments and the preliminary results in Section 2.1, we do notice **an accuracy improvement** for both pretraining and subnetwork training with a sufficient training recipe. However, the accuracy gap between pretrained network and subnetwork is still **non-negligible**. For instance, consider the case using ResNet-20 on CIFAR-10 at $s = 0.914$ in Figure 3, the Jackpot winning ticket is not identified, because the highest accuracy of the subnetwork by LT-IMP has a noticeable gap ($> 0.5\%$) compared to the highest pretraining accuracy. Take VGG-16 on CIFAR-10 at $s = 0.914$ as another example, although the subnetwork achieves similar accuracy with pretraining, there is no accuracy gap ($< 0.5\%$) between LT and RR, thus no tickets are found.

Recall the principles for identifying the winning ticket, all the cases are verified at the best suited learning rate, and please note that if there exists any non-trivial sparsity ratio (please check Appendix F.1 for results at all sparsity ratios) that makes the subnetwork meet the conditions, we call the Jackpot winning ticket exist for this network. Under the rigorous definition, the odds for getting a Jackpot winning ticket is low, but we believe the Jackpot winning ticket is likely to be existing in a network with an appropriate size and trained using a desirable learning rate (please check Appendix F.2 for more details). For instance, in Figure 3, the case of MobileNet-v1 on CIFAR-10 at $s = 0.832$ reaches the boundary of Jackpot winning ticket conditions, as the accuracy gaps between LT and pretraining, and between RR and LT are both around 0.5%.

**Do secondary prize tickets exist in our evaluation?**

Yes. secondary prize tickets exist in most of the networks on small datasets. Note that the "winning tickets" found in previous works are (at most) similar to the secondary prize tickets based on our definition. Again, we use ResNet-20 at $s = 0.914$ as an example. In Figure 3, secondary prize ticket

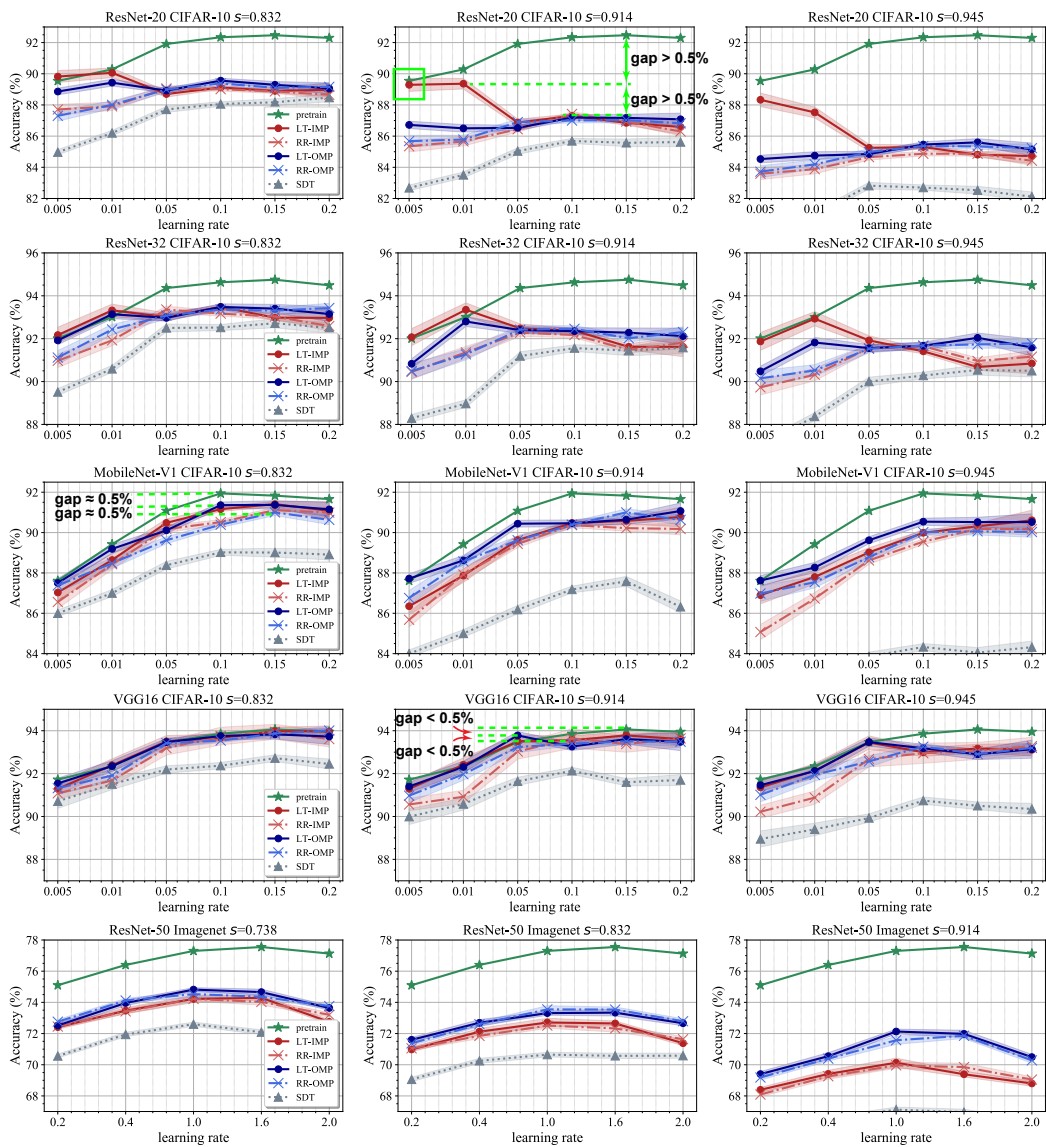

Figure 3: Lottery ticket experiments with different networks, datasets and (initial) learning rates. CIFAR-10 results are ordered by network size. ResNet-50 results on ImageNet-1K are also included.

exists in the green box, because the LT accuracy is similar with the pretraining accuracy at the same learning rate (0.005), while an accuracy gap ($> 0.5\%$) between LT and RR exists. However, the capacity of the network (in our cases, the number of weights in a network) determines the maximum sparsity at which a secondary prize ticket can be found. For instance, a relatively small network ResNet-20 can identify the secondary prize ticket at a maximum sparsity ratio of 0.914 on CIFAR-10, while larger networks such as ResNet-32, ResNet-18 and VGG-16 can identify secondary prize tickets on sparsity ratio of 0.945 or higher (refer to Appendix F.1). But on a medium and large-scale dataset as Tiny-ImageNet and ImageNet-1K, no clear secondary prize tickets are identified using ResNet-18 or ResNet-50. We believe a larger network may be able to identify one on ImageNet-1K.

**Which pruning method is better, `IMP`, `OMP` , or it does not matter?**

Comparing the results regarding network structures, we find that when residual connections exist in the network, `IMP` is more preferable than `OMP`, and when there are no residual connections the `IMP` has no advantages over `OMP`. To further investigate it with "apple-to-apple" comparison, we construct a "ResNet-32-like" network, by removing all residual connections from ResNet-32 while

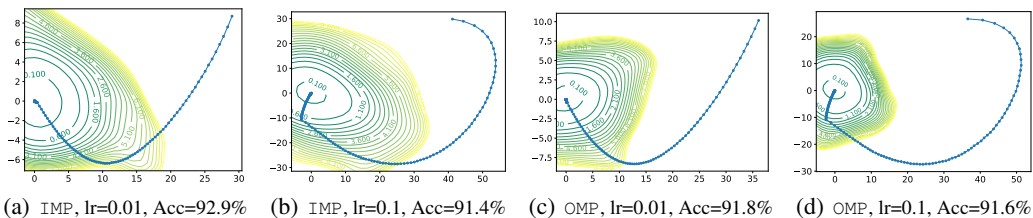

(a) IMP, lr=0.01, Acc=92.9%   (b) IMP, lr=0.1, Acc=91.4%   (c) OMP, lr=0.01, Acc=91.8%   (d) OMP, lr=0.1, Acc=91.6%

Figure 4: Training trajectories along the loss surface contours of ResNet-32 on CIFAR-10 at sparsity ratio of 0.945.

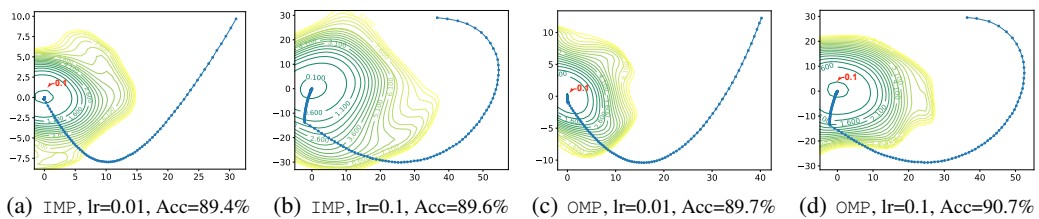

(a) IMP, lr=0.01, Acc=89.4%   (b) IMP, lr=0.1, Acc=89.6%   (c) OMP, lr=0.01, Acc=89.7%   (d) OMP, lr=0.1, Acc=90.7%

Figure 5: Training trajectories along the loss contours of ResNet-32-like network without residual connections on CIFAR-10 at sparsity ratio of 0.945.

leaving all else intact. We then evaluate the accuracy of IMP and OMP on ResNet-32, versus the newly constructed ResNet-32-like network. We also visualize both optimization trajectories along the contours of the loss surface, using the classical method in [21, 22].

According to Figure 4 that the residual connections exist in ResNet-32, a subnetwork using IMP explores a much smoother route than using OMP as its contour is smoother and close-to-convex (a larger landscape area with mild variance, and a larger basin in the middle of it [22]), which indicates that the optimization route may be smooth towards local minima.

When there are no residual connection as Figure 5 shows, however, we do not see much difference between IMP and OMP. Compare to the IMP method in Figure 4, the advantages of the IMP to OMP is diminished. Note that the landscape will become much more rugged if residual connections are removed from a network [22]. We conjecture that in our constructed no-residual ResNet-32, the optimization becomes too difficult and neither IMP nor OMP is effective enough to explore a smooth route towards local minima: hence no much difference observed between them.

**What learning rate is more likely to help identifying the winning tickets?**

We notice that when residual connections exist, the subnetwork achieves higher accuracy at a relatively small learning rate, while a larger learning rate is more preferable in training of a subnetwork without residual connections. In Figure 4, as the residual connection makes the landscape become much smoother [22], we can see a subnetwork trained with a small initial learning rate 0.01 achieves a larger contour and a larger basin in the middle, while the contour and basin area with large learning rate 0.1 are relatively small. We conjecture that the optimization is much easier for a smaller initial learning rate on a smooth loss surface, leading to a better network performance. Without residual connections (as Figure 5), the above observations are exactly the opposite. Note that the no-residual ResNet-32 creates a more rugged landscape, thus a small initial learning rate 0.01 is more likely to stuck in a sub-optimal local minima, while a large initial learning rate is unlikely to, therefore the SGD process is more likely to find a desired path to high quality solutions.

**When does $\theta_0$ benefit subnetwork training?**

We find that the secondary prize tickets are more likely to be found at a relatively small learning rate. To analyze the reason, we use a *correlation indicator* $R_p(\theta, \theta')$ to quantify the number of overlapped indices of the top-$p \cdot 100\%$ large-magnitude weights between two different sets of weights. We say the correlation between $\theta$ and $\theta'$ is weak if $R_p(\theta, \theta') \approx p$, and when $R_p(\theta, \theta') > p$, the correlation is positive. The detailed definition and explanation of the correlation indicator is shown in Appendix G. We evaluate the correlations between $(\theta_0 \odot m)$ and $(\theta_T \odot m)$, and between $(\theta'_0 \odot m)$ and $(\theta_T \odot m)$

regarding different learning rates on ResNet-20 and VGG-16 as Figure 6 shows. When using a relatively small learning rate, we find that the accuracy of $f(x; (\theta_0 \odot m)_T)$ is closer to pretraining accuracy than $f(x; (\theta'_0 \odot m)_T)$ does. In this case, the correlation between $(\theta_0 \odot m)$ and $(\theta_T \odot m)$ is positive while $(\theta'_0 \odot m)$ and $(\theta_T \odot m)$ is weak. When the correlation between $(\theta_0 \odot m)$ and $(\theta_T \odot m)$ is positive, the weights that are large in magnitude in pretraining network are likely to also be large in a trained subnetwork, thus a relatively close accuracy is observed. When the correlation does not exist, using $\theta_0$ or $\theta'_0$ in the subnetwork makes no difference to the final accuracy.

**Does the size of the dataset affects the patterns for the winning tickets identification?**

We find the patterns for the identified winning tickets are different on a relatively large-scale dataset, such as Tiny-ImageNet and ImageNet-1K. For all the ResNet architectures we evaluate, `OMP` outperforms `IMP`, and small learning rates are not preferable in training a subnetwork. We provide more discussion in Appendix F.3.

**Does weight rewinding improve the accuracy?**

We find the weight rewinding technique [5] consistently improves the subnetwork accuracy. We generalize the weight rewinding technique into the winning ticket identification principles, and perform a series of experiments. Due to space limits, the results are discussed in Appendix D.

Figure 6: Correlation between weights in subnetwork and pretrained network with different learning rates. The subnetwork we use has $s = 0.832$ and we set $p = 0.1$.

### 3.2 How to Quickly Win a Prize in a Lottery Game – A Guideline

In this section, we summarize the patterns we find through the extensive experimental results, and present in the form of a guideline to help quickly identify the *Jackpot winning ticket* and *secondary prize ticket* (both referred as *ticket* below for simplicity). Our guideline is presented as follows:

1. On a small dataset using networks with residual connections, `IMP` is better than `OMP`. When the network has no residual connections, `IMP` has no advantages over `OMP`.

2. On a small dataset using networks with residual connections, the subnetwork prefers a relatively small learning rate to find the tickets. When the network has no residual connections, small learning rate is not preferable.

3. When the network is redundant (e.g., a large network on a small-scale dataset), the maximum sparsity that a ticket can be found is relatively high, and vice versa.

4. When the (sub)network prefers large learning rates, using different initialization yields the similar accuracy in subnetwork training.

## 4 Ablation Study on Subnetwork Training with Different Learning Rates

In the lottery ticket hypothesis studies, it is a standard setting to use the same learning rate in pretraining (for finding the mask by pruning thereafter) and subnetwork training (for training the sparse model) [1, 5, 12]. In this paper, for each learning rate we have evaluated, the pretraining and subnetwork training also adopt the same learning rate setting. However, it does not consider the possibility that a subnetwork may prefer a different learning rate than it is used in pretraining. One key observation in [23] suggests that it is desirable to use different learning rates during pretraining and subnetwork training, and that doing so may lead to the well-performing lottery tickets.

According to our principle ④ and ⑤ for identifying winning tickets, any learning rate that satisfying the conditions would make a successful Jackpot winning ticket or secondary prize ticket. Therefore, the rigorous definition of lottery ticket hypothesis and the principles for identifying winning tickets are valid (when consider different combinations of learning rates) and can hold *true* for future research. In Table 4, we evaluate two series of experiments with two different pretraining learning rates using ResNet-20. We find that using different learning rates in pretraining and subnetwork training slightly

Table 4: Ablation results using ResNet-20 on CIFAR-10 at sparsity 0.914. The shaded area indicates the learning rate that finds the better subnetwork accuracy.

| Pretraining lr (Acc %): 0.01 (90.3) | | | Pretraining lr (Acc %): 0.1 (92.4) | | |
|---|---|---|---|---|---|
| LT lr | `IMP` Acc (%) | `OMP` Acc (%) | LT lr | `IMP` Acc (%) | `OMP` Acc (%) |
| 0.001 | 87.5 | 83.3 | 0.01 | 85.3 | 85.8 |
| 0.005 | **89.7** | 85.3 | 0.05 | 86.6 | **87.4** |
| 0.01 | 89.4 | 86.5 | 0.1 | 87.3 | 87.2 |
| 0.05 | 87.9 | 87.2 | 0.15 | 86.7 | 87.3 |

benefits the accuracy (e.g., 89.7% vs. 89.4% `IMP` accuracy in the case of pretraining using learning rate of 0.01, or 87.4% vs. 87.2% `OMP` accuracy in the case of pretraining using learning rate of 0.1) but is not changing our previous observations. The results further strengthen our claim that the Jackpot winning ticket might exist in a network when trained using a desirable learning rate.

## 5 Related Works

**Lottery Ticket Hypothesis.** The lottery ticket hypothesis and the definition of the "winning ticket" are firstly proposed in [1]. Concurrent work [9] finds that the identical initialized weights will not provide any advantage over training with randomly initialized weights at relatively large learning rates. Later works [9, 24] also confirm that the matching subnetworks at nontrivial sparsity are hard to find in more challenging tasks. The following works [5, 12] extend the subnetwork training from initial weights to the weights at early stage of pretraining (rewinding), and improve the accuracy in more challenging tasks at nontrivial sparsity. Concretely, [12] makes a key observation that subnetworks are stable to SGD noise in early stage of training, which explains why rewinding technique succeeds in LTH. In this paper, we recognize rewinding technique as a successful approach to achieve dense network accuracy for the subnetworks, but our study focus on the effects and their rationales of different network characteristics and the experimental conditions in LTH.

Besides computer vision tasks, the lottery ticket hypothesis is also investigated in many other tasks [24, 25, 6, 7, 26, 27, 28, 29, 30, 31]. Other works [6, 32] further extend the lottery ticket hypothesis to a pre-trained BERT model. On object detection task, [33] proposes a guidance to find task-specific winning tickets for object detection, instance segmentation, and keypoint estimation. [34, 35] have studied the lottery ticket hypothesis in unsupervised learning to reveal how well the tickets are transformed between different datasets.

**Find Winning Ticket at Early Stage of Training.** The potential of training a sparse network from initialization suggested by the lottery ticket hypothesis has motivated the study of deriving the "winning tickets" at an early stage of training, thereby accelerating training process. There is a number of work in this direction. [4, 36] conduct a retraining process after searching sub-network topology for a few epochs. [37] examines the network state during early iterations of training, and analyzes the weight distribution and its reliance on the dataset. SNIP [38] finds the sparse mask based on the saliency score of each weight that is obtained after training the dense model for only a few iterations. GraSP [39] prunes weights based on preserving the gradient flow in the network.

## 6 Conclusion and Discussion of Broader Impact

In this paper, we investigate the underlying condition and rationale behind the lottery ticket hypothesis. By revisiting the original definition, we find out that the current controversies over this topic is largely related to the quality of the training recipe. We propose a rigorous definition of the lottery ticket hypothesis, as well as the principles for identifying the true "Jackpot winning ticket" or "secondary prize ticket". We perform sanity checks for the lottery tickets through extensive experiments over multiple deep models on different datasets, and empirically study the patterns we observe by quantitative analysis. Meanwhile, we develop a guideline based on our summarized patterns, which potentially facilitates the research process on the topic of the lottery ticket hypothesis. The research is scientific in nature and we do not envision it to generate any negative societal impact.

## Acknowledgment

This work is partly supported by the National Science Foundation CCF-1919117 and ECCS-2053272, and Army Research Office/Army Research Laboratory via grant W911NF-20-1-0167 (YIP) to Northeastern University. Any opinions, findings, and conclusions or recommendations expressed in this material are those of the authors and do not necessarily reflect the views of NSF or ARO.

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
