# Appendix

## A    Preliminary Results on Different Training Recipes

As a supplement to Figure 1, we plot the accuracy vs. sparsity curves in Figure A.1 which include all sparsity levels for `IMP` process. We also train the LT-`OMP` with the same sparsity levels as `IMP`. Corresponding RR-`IMP` and RR-`OMP` are all included.

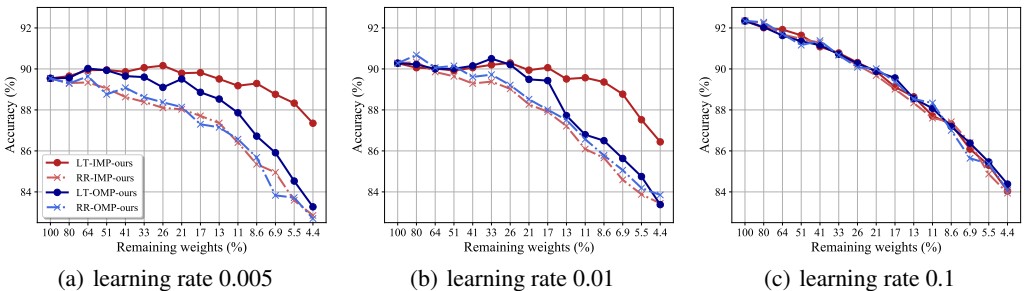

(a) learning rate 0.005       (b) learning rate 0.01       (c) learning rate 0.1

Figure A.1: Full results for all sparsity ratios of ResNet-20 on CIFAR-10 dataset with different learning rates. We can clearly notice that the dense models (i.e., with 100% remaining weights) have different accuracy. At learning rates of 0.005 and 0.01, the dense model accuracy are around 90% and the LT-`IMP`/`OMP` maintain the accuracy of dense models while keeping clear advantages over RR-`IMP`/`OMP`. At learning rate of 0.1, accuracy of dense model is over 92% and the LT-`IMP`/`OMP` accuracy drops immediately after the models are pruned. Additionally, no clear accuracy gap between LT and RR results.

## B    Lottery Ticket Hypothesis – A Rigorous Definition with Weight Rewinding

The weight rewinding technique [5] finds that the accuracy of the subnetwork improves significantly when using the weights at early stage of pretraining (i.e. $\theta_t$ where $t < T$) as the initial point to train the subnetwork $f(x; (\theta_t \odot m))$. Since the weights have rewound to an early stage of pretraining, the training epochs for subnetworks will decrease by $t$ to ensure same training efforts (i.e. training subnetwork for $T - t$ epochs and arrives at $f(x; (\theta_t \odot m)_{T-t})$). We define the following settings regarding the weight rewinding technique:

- *Weight rewinding with `OMP` (WR-`OMP`):* We directly apply mask $m_O$ to initial weights $\theta_t$, resulting in weights $\theta_t \odot m_O$ and network function $f(x; \theta_t \odot m_O)$.

- *Weight rewinding with `IMP` (WR-`IMP`):* We apply $m_I$ to initial weights $\theta_t$, and get $f(x; \theta_t \odot m_I)$.

We generalize the weight rewinding technique into our rigorous definition of the lottery ticket hypothesis as follows:

**The lottery ticket hypothesis – a rigorous definition with weight rewinding.** *Under a non-trivial sparsity ratio, there exists a subnetwork that – when rewinds to initial or early stage of the pretraining weights and trained in isolation with a decent learning rate – can reach similar accuracy with the well-trained original network using the same or fewer iterations, while showing clear advantage in accuracy compared to a randomly reinitialized subnetwork as well as an equivalently parameterized small-dense network.*

**The principles for the identification of the rewinding winning tickets.** Similar with the principles for identification of the winning tickets in our main paper, we list the conditions for identifying winning ticket by rewinding technique as follows:

① A non-trivial sparsity ratio $s$ and a sufficient training epochs $T$ are adopted for the subnetwork.

② SDT of $f(x; \theta_T^{SD})$ shows clear accuracy drop compared to the well-trained subnetwork.

③ There exists a learning rate such that the subnetwork $f(x; (\theta_t \odot m)_{T-t})$ achieves notably higher accuracy (with a clear gap) than $f(x; (\theta'_0 \odot m)_T)$ trained with any learning rates.

④ There exists a learning rate such that the subnetwork $f(x; (\theta_t \odot m)_{T-t})$ achieves accuracy similar to or higher than the pretrained network $f(x; \theta_T)$ at the same learning rate.

⑤ There exists a learning rate such that the subnetwork $f(x; (\theta_t \odot m)_{T-t})$ achieves accuracy similar to or higher than the *well-trained* original network $f(x; \theta_T)$ (i.e., trained with an appropriate learning rate and sufficient number of training epochs).

We also need to point out that the weight rewinding technique is fundamentally different from the essential research purpose of the lottery ticket hypothesis. The study of winning tickets (also referred to as rewinding to $\theta_0$) explores the network initial states and topology, while rewinding technique investigates at what pretraining stage $t < T$ does the subnetwork $f(x; (\theta_t \odot m)_{T-t})$ achieve similar accuracy with pretraining. Practically, the lottery ticket hypothesis provides potential possibility for sparse training at initialization, but with weight rewinding, dense network training is required that is not memory-efficient.

## C   A Mathematical Version of the Rigorous Definition of the Lottery Ticket Hypothesis

In this section, we formulate our rigorous definition of the lottery ticket hypothesis proposed in Section 2.2 in a mathematical representation.

---

**The lottery ticket hypothesis – a rigorous definition.** *Suppose that there is a sub-network $f(x; m \odot \theta_0)$ in which the sparse mask $m \in \{0, 1\}^{|\theta|}$ under a non-trivial sparsity ratio that is acquired from a certain pruning algorithm and is associated with the initial weights $\theta_0$. After $T$-epoch training, let $A_{\mathrm{LT}}$ be the test accuracy achieved by $f(x; m \odot \theta_0)$. Moreover, let $A_{\mathrm{PRE}}$ denote the accuracy of the pretrained dense network from $f(x; \theta_0)$ in a sufficient $T$-epoch training with a decent learning rate. Associated with $f(x; m \odot \theta_0)$, let $f(x; \theta^{SD})$ and $f(x; \theta'_0 \odot m)$ denote a small-dense network with model size the same as $\|m\|$ and a randomly reinitialized subnetwork $f(x; \theta'_0 \odot m)$, with accuracies $A_{\mathrm{SD}}$ and $A_{\mathrm{RR}}$, respectively. **The lottery ticket hypothesis is then stated below:** $\exists\, f(x; \theta_0 \odot m)$, when trained with $T' \leq T$ epochs, can reach to the accuracy $A_{\mathrm{LT}}$ satisfied with $A_{\mathrm{LT}} \approx A_{\mathrm{PRE}}$, $A_{\mathrm{LT}} > A_{\mathrm{SD}}$, $A_{\mathrm{LT}} > A_{\mathrm{RR}}$, where $>$ indicates a clear accuracy gap.*

---

## D   Experimental Results of Weight Rewinding.

In this section, we show the results of weight rewinding technique for subnetwork training. We also plot the original lottery ticket results (i.e. rewind to $\theta_0$) along with the weight rewinding results to demonstrate the accuracy improvement.

Figure D.1 – D.14 demonstrate all weight rewinding results based on the networks and datasets we evaluated in Table 2. Specifically, for the subnetwork training, we rewind to the pretraining weights at approximate 5% of the total training epochs.

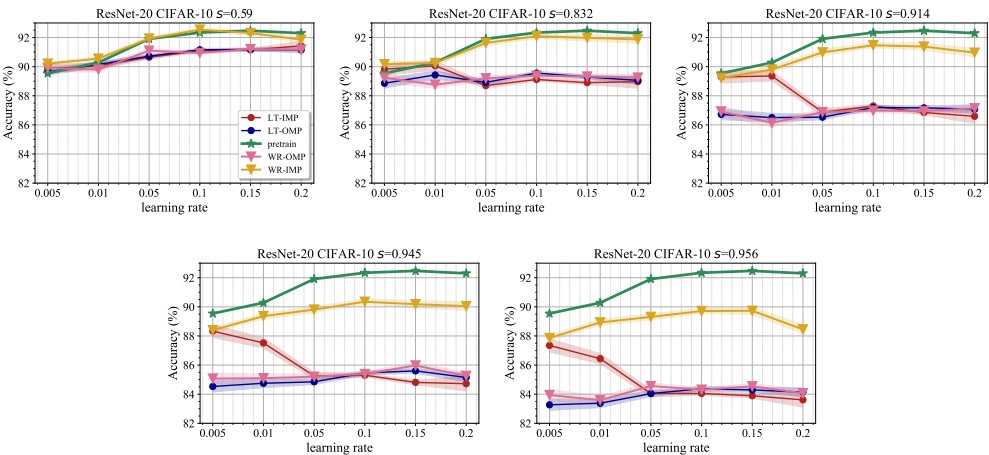

Figure D.1: Weight rewinding experiments with ResNet-20 on CIFAR-10 at different sparsity ratios.

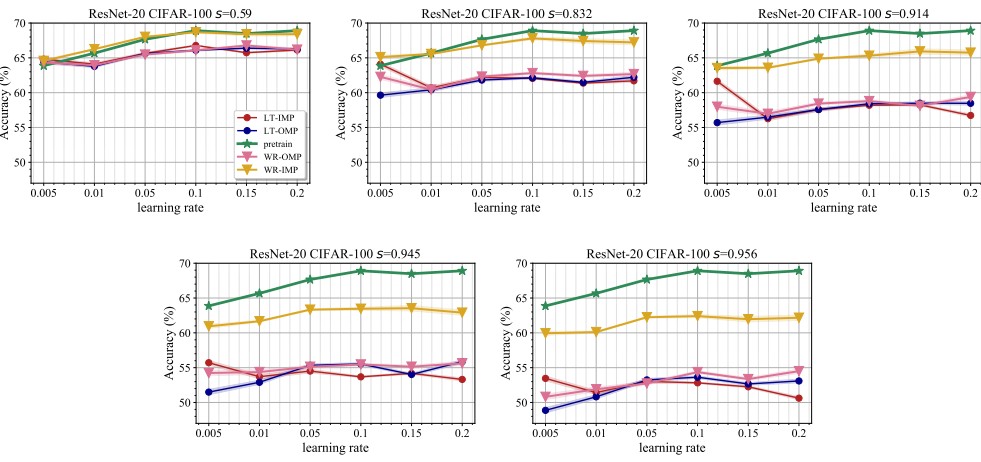

Figure D.2: Weight rewinding experiments with ResNet-20 on CIFAR-100 at different sparsity ratios.

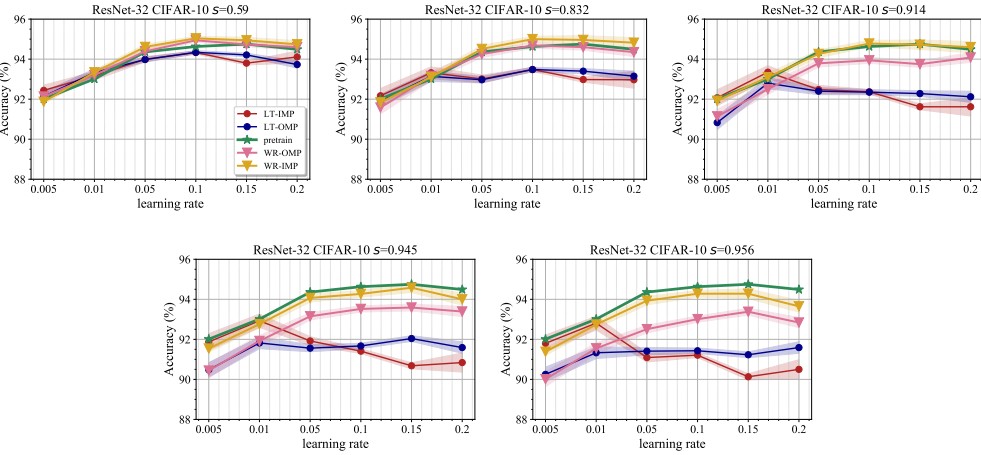

Figure D.3: Weight rewinding experiments with ResNet-32 on CIFAR-10 at different sparsity ratios.

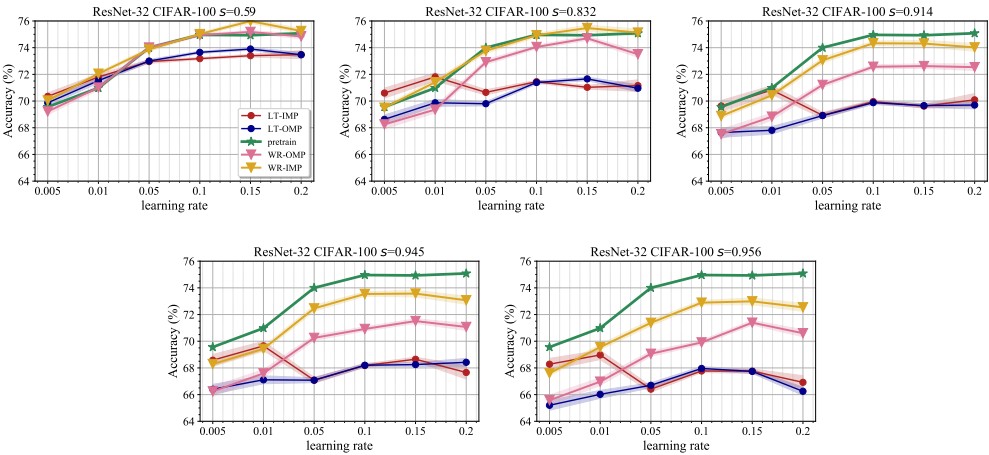

Figure D.4: Weight rewinding experiments with ResNet-32 on CIFAR-100 at different sparsity ratios.

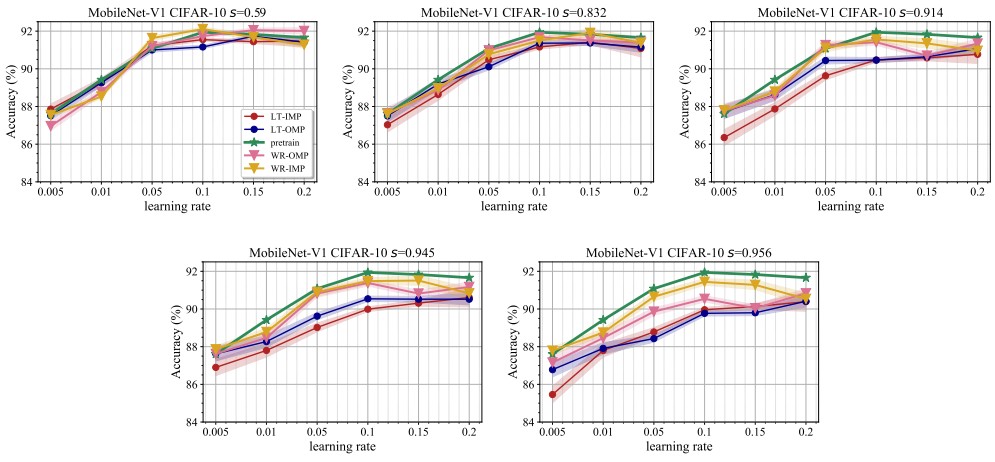

Figure D.5: Weight rewinding experiments with MobileNet-v1 on CIFAR-10 at different sparsity ratios.

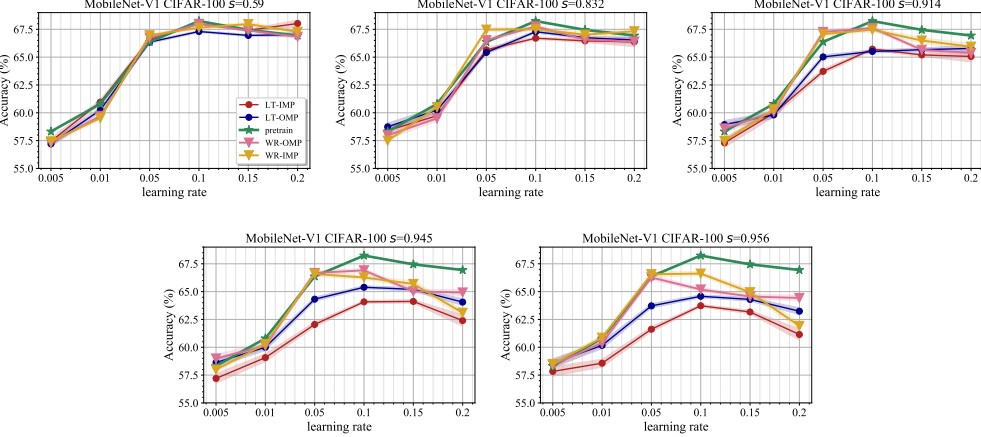

Figure D.6: Weight rewinding experiments with MobileNet-v1 on CIFAR-100 at different sparsity ratios.

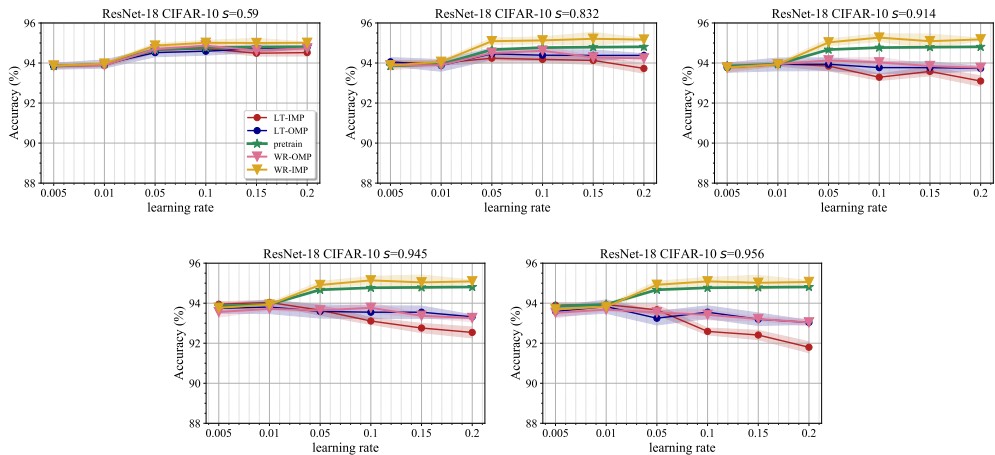

Figure D.7: Weight rewinding experiments with ResNet-18 on CIFAR-10 at different sparsity ratios.

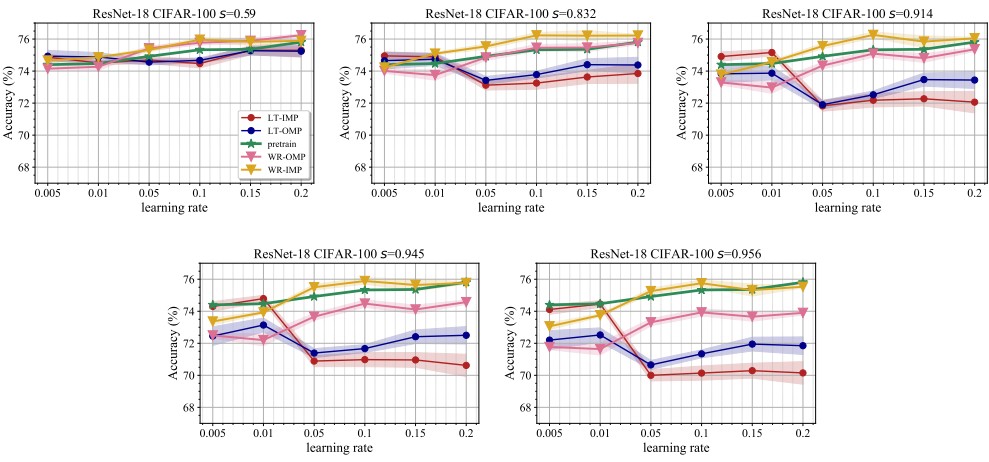

Figure D.8: Weight rewinding experiments with ResNet-18 on CIFAR-100 at different sparsity ratios.

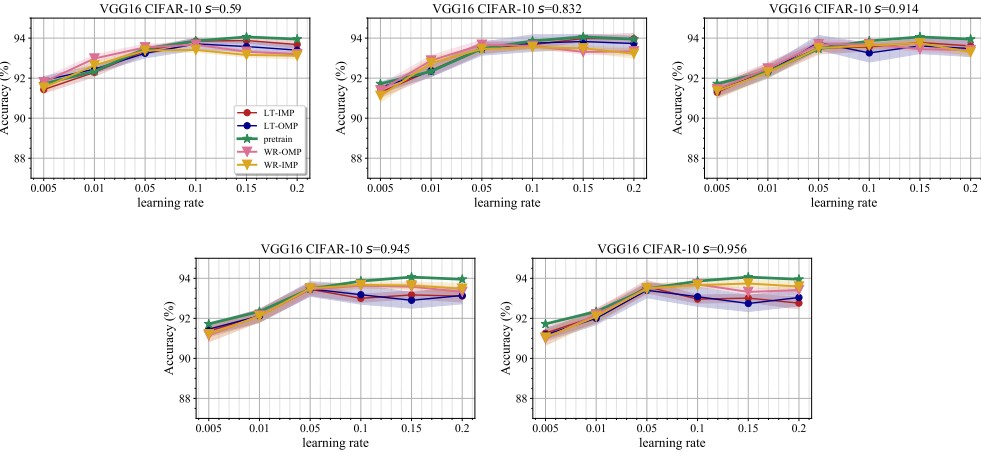

Figure D.9: Weight rewinding experiments with VGG-16 on CIFAR-10 at different sparsity ratios.

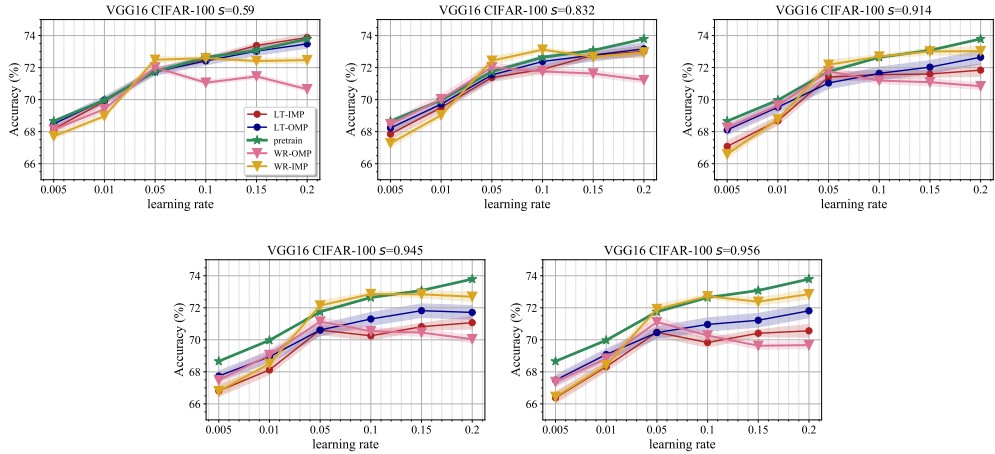

Figure D.10: Weight rewinding experiments with VGG-16 on CIFAR-100 at different sparsity ratios.

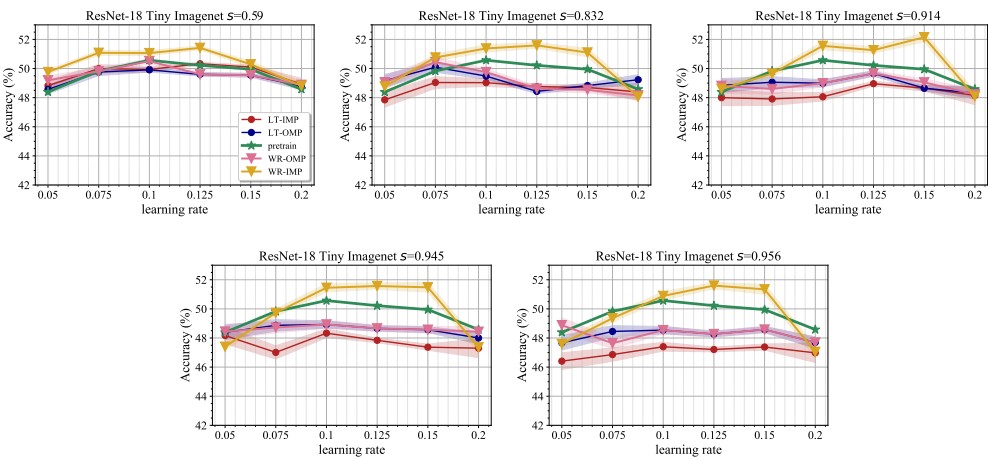

Figure D.11: Weight rewinding experiments with ResNet-18 on Tiny-ImageNet at different sparsity ratios.

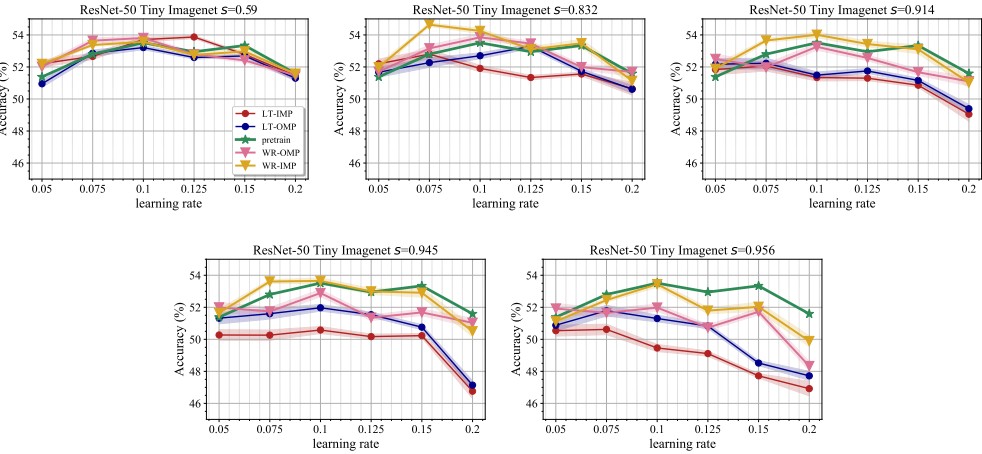

Figure D.12: Weight rewinding experiments with ResNet-50 on Tiny-ImageNet at different sparsity ratios.

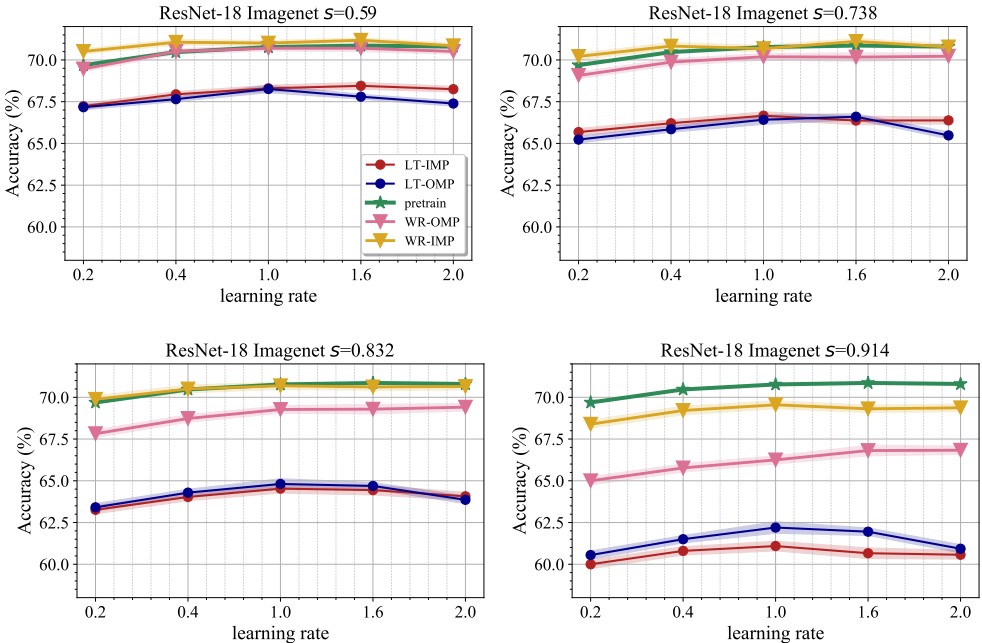

Figure D.13: Weight rewinding experiments with ResNet-18 on ImageNet-1K at different sparsity ratios.

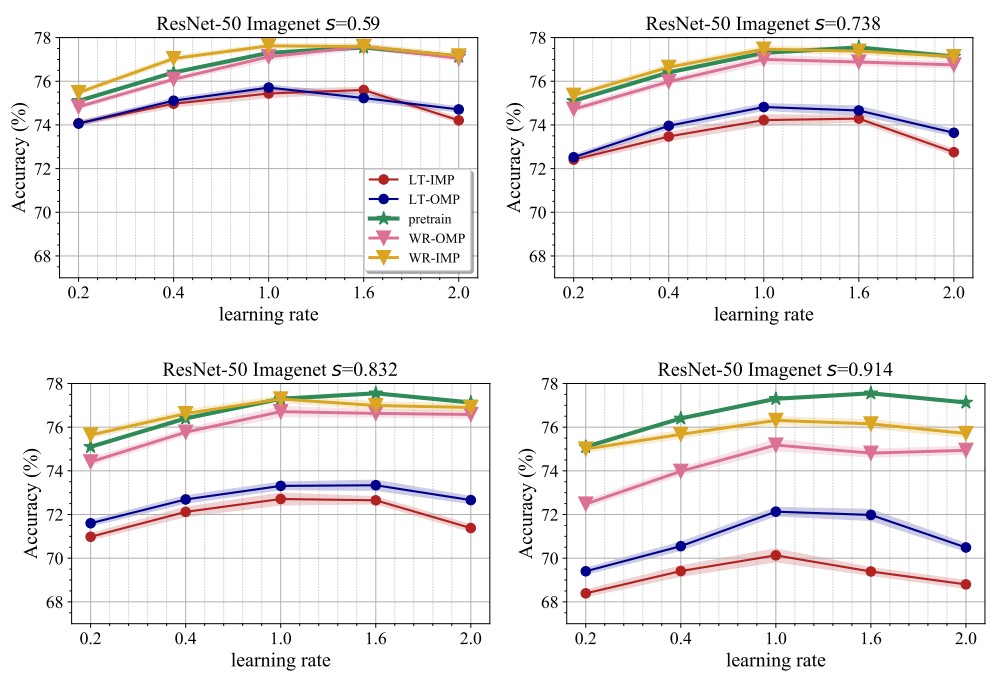

Figure D.14: Weight rewinding experiments with ResNet-50 on ImageNet-1K at different sparsity ratios.

# E   Experiment Setups.

We list the hyperparameter settings for our experiments in Table E.1.

Table E.1: Hyperparameter settings.

| Experiments | CIFAR-10/100 | Tiny-ImageNet | ImageNet |
|---|---|---|---|
| Training epochs ($T$) | 160 | 160 | 90 |
| Rewinding epochs ($t$) | 8 | 8 | 5 |
| Batch size | 64 | 32 | 1024 |
| Learning rate scheduler | step | step | cosine |
| Learning rate decay (epoch) | 80-120 | 80-120 | n/a |
| Learning rate decay factor | 10 | 10 | n/a |
| Momentum | 0.9 | 0.9 | 0.875 |
| $\ell_2$ regularization | 5e-4 | 5e-4 | 3.05e-5 |
| Warmup epochs | 0 (75 for VGG-16) | 20 | 8 |
| IMP prune ratio (per iteration) IMP total iterations | 20% 14 | 20% 14 | 20% 11 |

# F Sanity Checks for Lottery Tickets: Full Results

## F.1 Full Evaluation Results

Figure F.1 – F.14 demonstrate the lottery ticket experiment results based on the networks and datasets we evaluated in Table 2. Specifically, we include the results that are not shown in the main paper.

## F.2 How Network Size Affects the Identification of the Jackpot Winning Tickets

From the experimental results on lottery ticket hypothesis, we find that the size of the network is a key factor for the identification of the Jackpot winning tickets. According to Table 2 and the results summary in Table 3, we conjecture that the degree of the over-parameterization of a network is highly related to whether Jackpot winning tickets exist. To find Jackpot winning tickets, the network size should be appropriate and a sufficient training recipe should be adopted. If a network is extremely under-parameterized (i.e., a very small network on a relatively large dataset), then it is unlikely to find a Jackpot winning ticket (please refer to the cases of ResNet-20 on CIFAR-10/100 in Figure 3 and Figure F.2). The reason is that the network capacity of the original dense network is already quite limited, thus the subnetwork, with even fewer parameters, are more likely to have even worse performance. As a result, the accuracy of the subnetwork is very unlikely to reach or close to the pretraining accuracy. On the other hand, if a network is extremely over-parameterized (i.e., a very redundant network on a relatively small dataset), then it is unlikely to find a Jackpot winning ticket (please refer to the cases of VGG-16 on CIFAR-10/100 in Figure 3 and Figure F.10). We believe the reason is that the capacity of the original network is too large, such that there is no difference using original initialization $\theta_0$ or random reinitialization $\theta'_0$ when training a subnetwork. When the size of the network is appropriate, Jackpot winning ticket is likely to be found or at least reach the boundary of the Jackpot winning conditions. For instance, the cases of MobileNet-v1 on CIFAR-10 in Figure 3 and ResNet-18 on CIFAR-100 in Figure F.8 find the Jackpot winning tickets at boundary condition. For CIFAR-10, a MobileNet-v1 is not too large nor small, thus the Jackpot winning ticket is likely to be found. For CIFAR-100, the dataset size is the same with CIFAR-10 but the classification task is more complicated. In this case, a larger ResNet-18 is suitable for finding the Jackpot winning ticket (or reach the boundary of it), while a VGG-16 is too large for identifying one.

## F.3 How Dataset Size Affects the Identification of the Winning Tickets

The experiment results show that when dataset size increases, the patterns for the identified winning tickets are different. On Tiny-ImageNet and ImageNet-1K, `OMP` outperforms `IMP` on all ResNet architectures we evaluate. While the underlying reasons are still remaining mysterious, we intuitively explain the reason: the current networks we use on Tiny-ImageNet and ImageNet-1K may not be able to fully represent rich features in the dataset, thus a more chaotic loss landscape. `IMP` and small learning rates, both representing a "small step" towards the objective, may be easily stucking in a sub-optimal local minima, while `OMP` and large learning rates are unlikely since they are more aggressive in pruning and optimization.

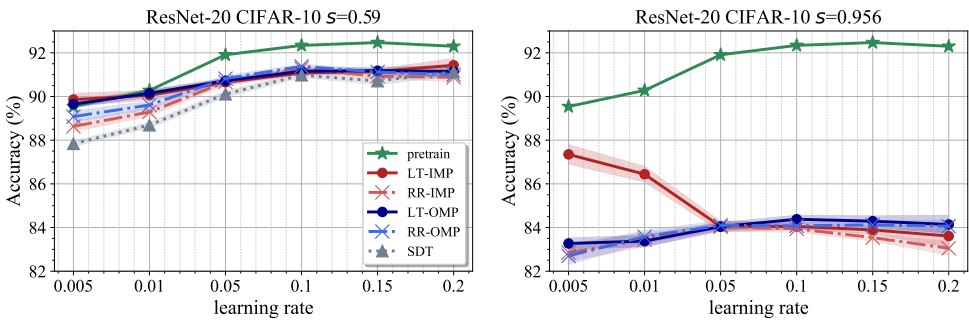

Figure F.1: Supplemental lottery ticket experiments with ResNet-20 on CIFAR-10 at sparsity ratio $s = 0.59$ and $s = 0.956$. Results of other sparsity ratios are shown in the main paper.

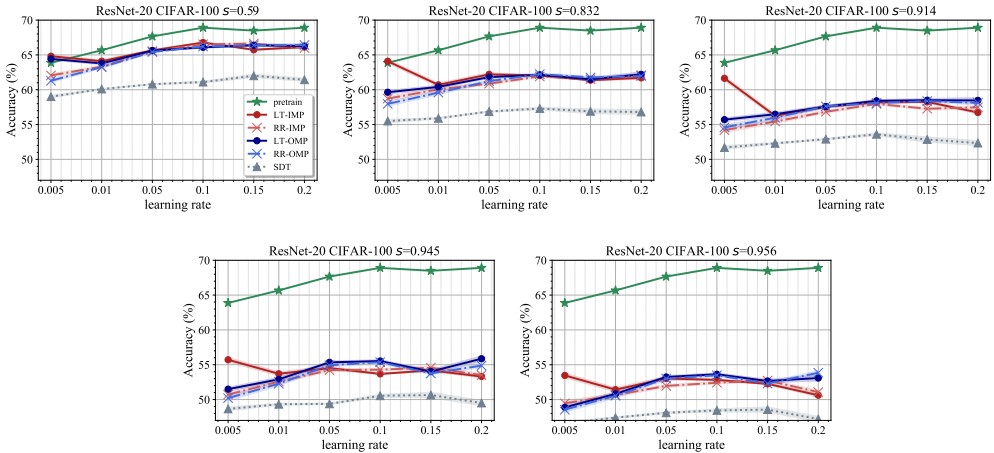

Figure F.2: Lottery ticket experiments with ResNet-20 on CIFAR-100 at different sparsity ratios.

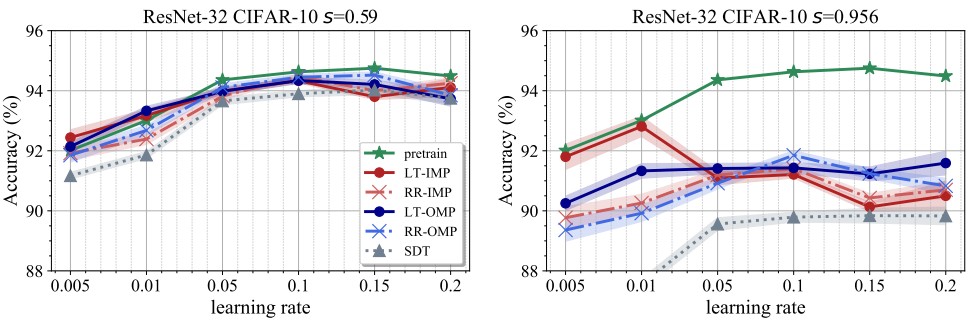

Figure F.3: Supplemental lottery ticket experiments with ResNet-32 on CIFAR-10 at sparsity ratio $s = 0.59$ and $s = 0.956$. Results of other sparsity ratios are shown in the main paper.

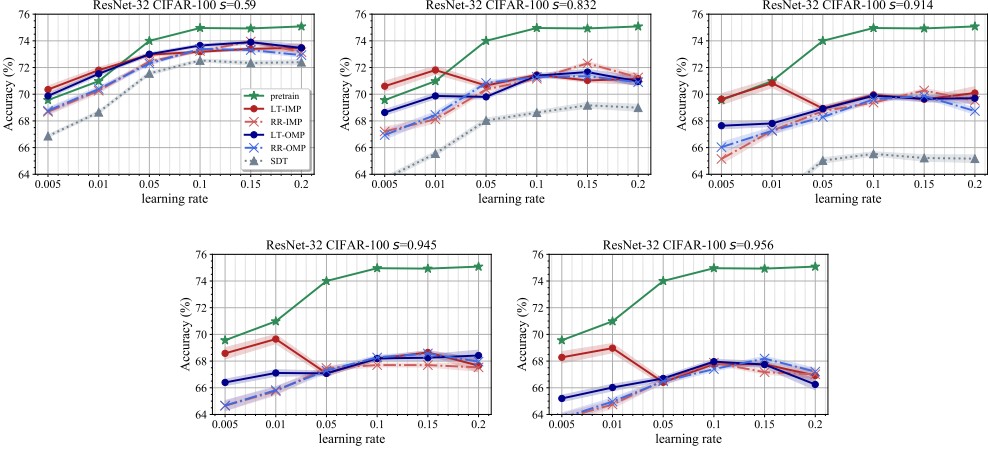

Figure F.4: Lottery ticket experiments with ResNet-32 on CIFAR-100 at different sparsity ratios.

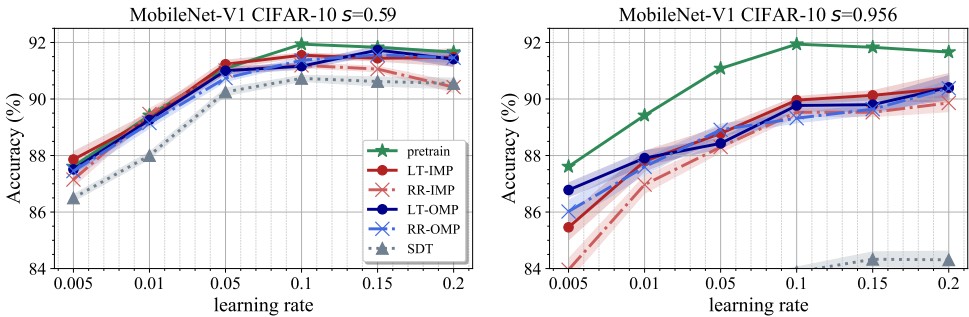

Figure F.5: Supplemental lottery ticket experiments with MobileNet-v1 on CIFAR-10 at sparsity ratio $s = 0.59$ and $s = 0.956$. Results of other sparsity ratios are shown in the main paper.

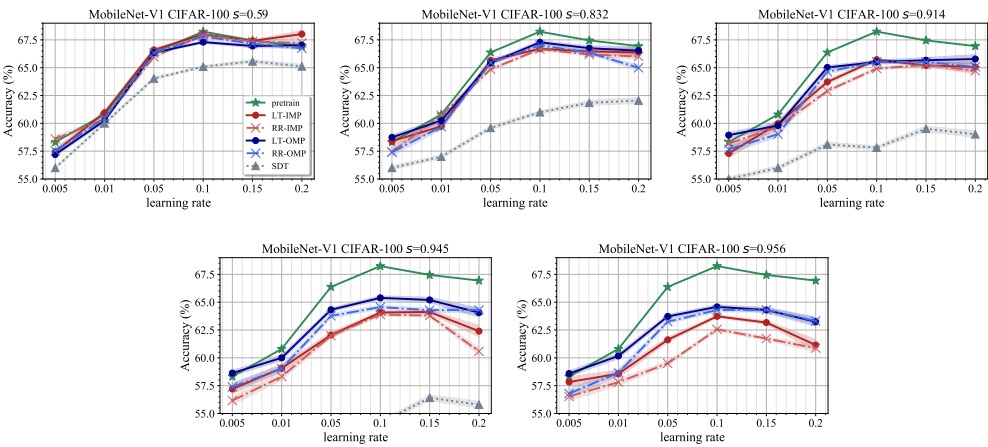

Figure F.6: Lottery ticket experiments with MobileNet-v1 on CIFAR-100 at different sparsity ratios.

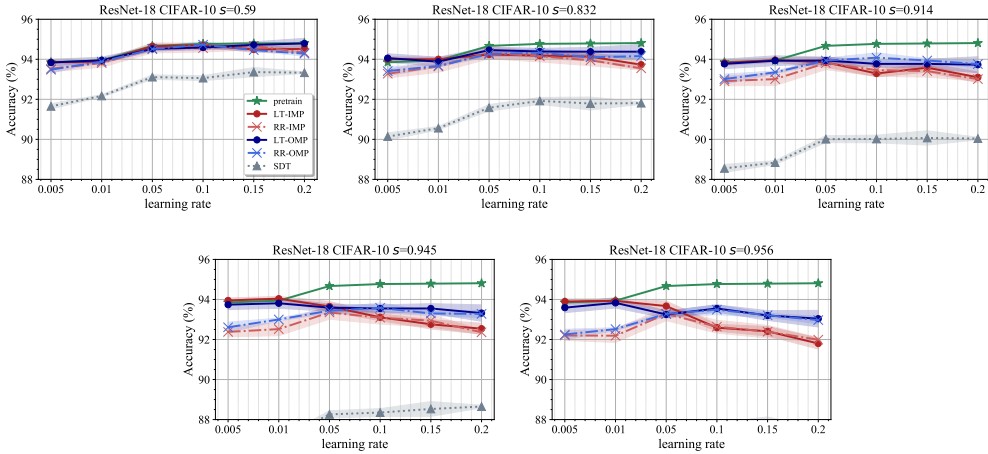

Figure F.7: Lottery ticket experiments with ResNet-18 on CIFAR-10 at different sparsity ratios.

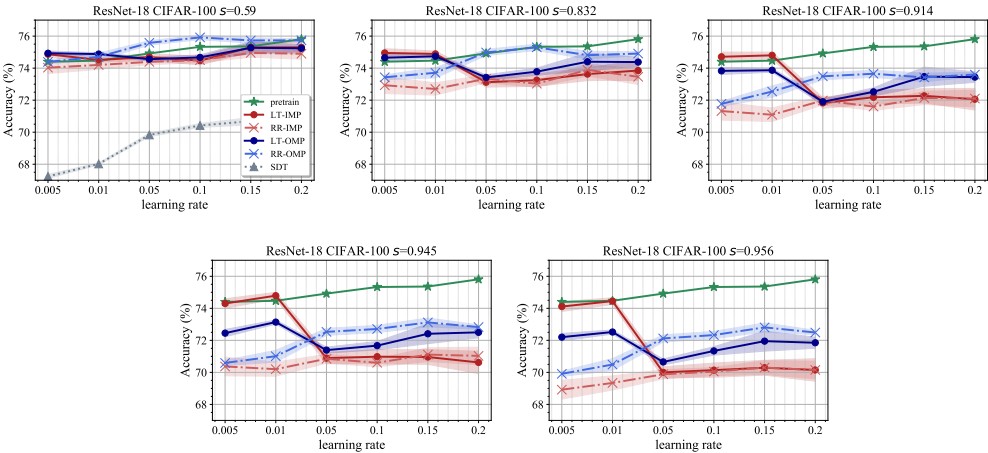

Figure F.8: Lottery ticket experiments with ResNet-18 on CIFAR-100 at different sparsity ratios.

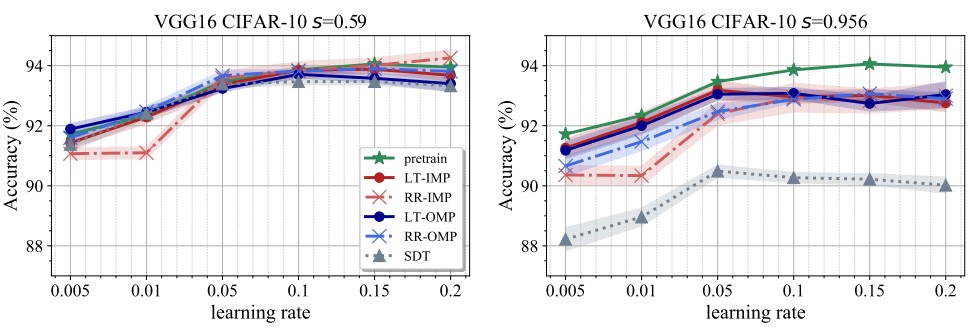

Figure F.9: Supplemental lottery ticket experiments with VGG-16 on CIFAR-10 at sparsity ratio $s = 0.59$ and $s = 0.956$. Results of other sparsity ratios are shown in the main paper.

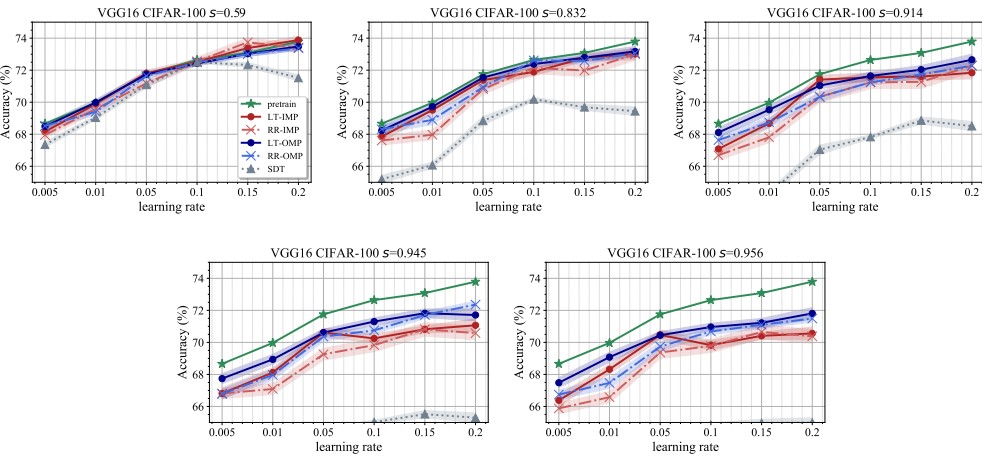

Figure F.10: Lottery ticket experiments with VGG-16 on CIFAR-100 at different sparsity ratios.

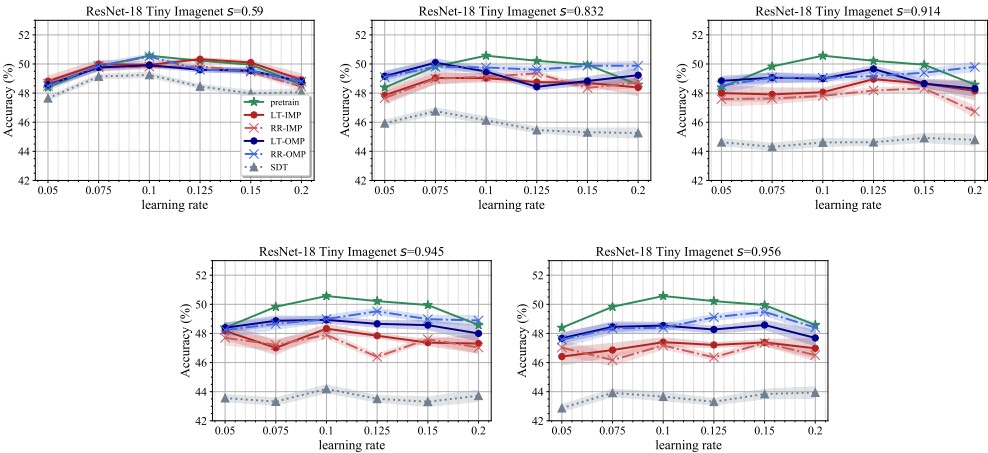

Figure F.11: Lottery ticket experiments with ResNet-18 on Tiny-ImageNet at different sparsity ratios.

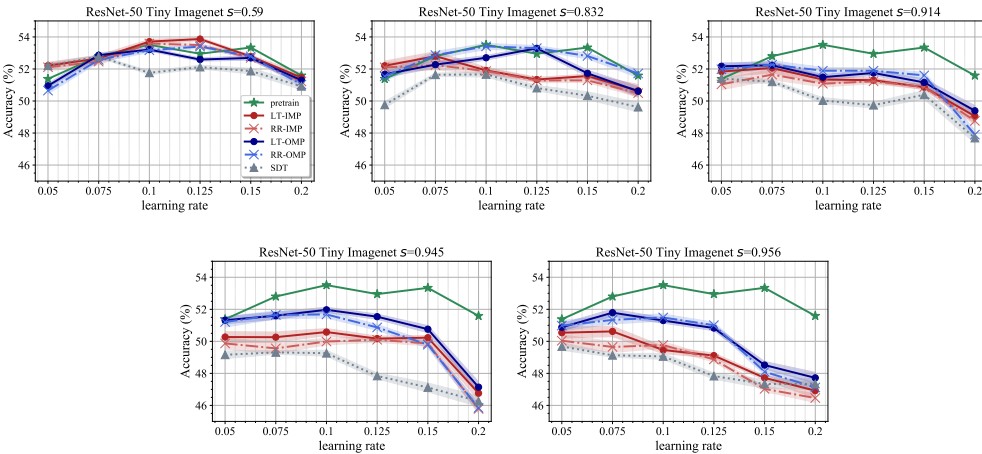

Figure F.12: Lottery ticket experiments with ResNet-50 on Tiny-ImageNet at different sparsity ratios.

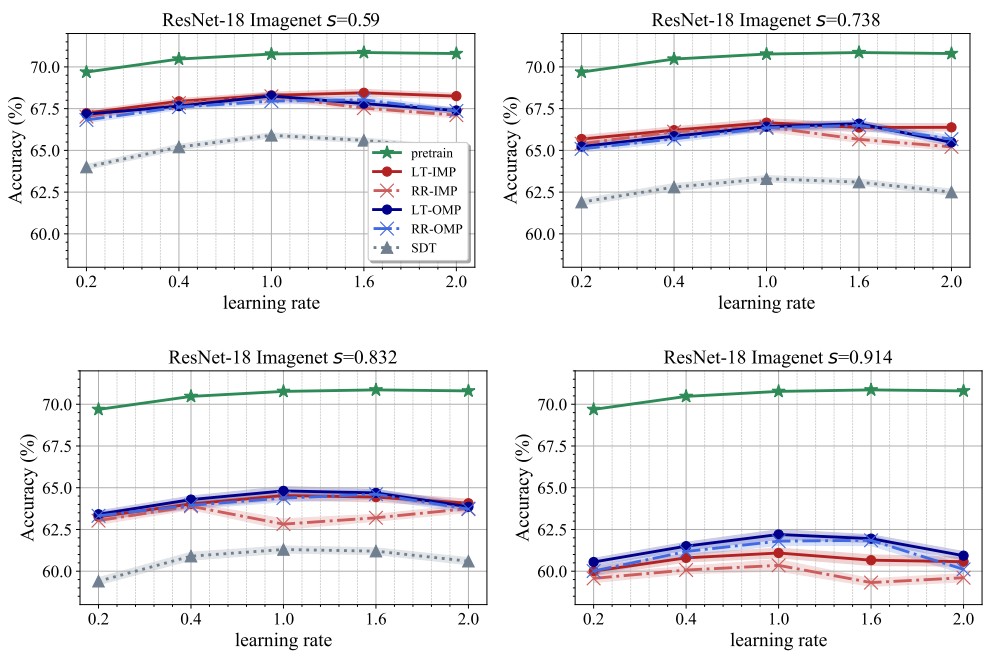

Figure F.13: Lottery ticket experiments with ResNet-18 on ImageNet-1K at different sparsity ratios.

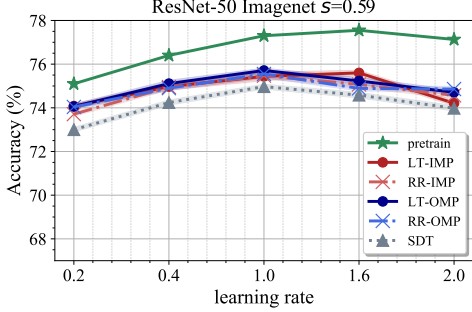

Figure F.14: Supplemental lottery ticket experiments with ResNet-50 on ImageNet-1K at sparsity ratio $s = 0.59$. Results of other sparsity ratios are shown in the main paper.

# G Correlation Indicator

Consider a DNN with two collections of weights $\theta$ and $\theta'$. Note that this is a general definition that applies to both the original DNN and sparse DNN (when the mask $m$ is applied and a portion of weights eliminated). We define the *correlation indicator* to quantify the amount of overlapped indices of large-magnitude weights between $\theta$ and $\theta'$. More specifically, given a DNN with $L$ layers, where the $l$-th layer has $N_l$ weights, the *weight index set* $T_p((\theta)^l)$ ($p \in [0,1]$) is the top-$p \cdot 100\%$ largest-magnitude weights in the $l$-layer. Similarly, we define $T_p((\theta')^l)$. Please note that for a sparse DNN, the portion $p$ is defined with respect to the number of remaining weights in the sparse (sub)network[3]. The intersection of these two sets includes those weights that are large (top-$p \cdot 100\%$) in magnitude in both $\theta$ and $\theta'$, and $\mathbf{card}\Big(T_p((\theta)^l) \cap T_p((\theta')^l)\Big)$ denotes the number of such weights in layer $l$. The correlation indicator (overlap ratio) between $\theta$ and $\theta'$ is finally defined as:

$$R_p(\theta, \theta') = \frac{\sum_l \mathbf{card}\Big(T_p((\theta)^l) \cap T_p((\theta')^l)\Big)}{p \cdot \sum_l N_l} \tag{1}$$

When $R_p(\theta, \theta') \approx p$, the top-$p \cdot 100\%$ largest-magnitude weights in $\theta$ and $\theta'$ are largely independent. In this case the correlation is relatively weak[4]. On the other hand, if there is a large deviation of $R_p(\theta, \theta')$ from $p$, then there is a strong correlation. Especially when $R_p(\theta, \theta') > p$, the weights that are large in magnitude in $\theta$ are likely to also be large in $\theta'$, indicating a positive correlation. Otherwise, when $R_p(\theta, \theta') < p$, it implies a negative correlation.

As shown in Figure G.1, the above correlation indicator will be utilized to quantify the correlation between a dense DNN and a dense DNN, i.e., $R_p(\theta_0, \theta_T)$ for DNN pre-training, and between a sparse DNN and a sparse DNN, i.e., $R_p(\theta_0 \odot m, \theta_T \odot m)$ and $R_p(\theta'_0 \odot m, \theta_T \odot m)$ for the cases of original initialization and random reinitialization under lottery ticket setting.

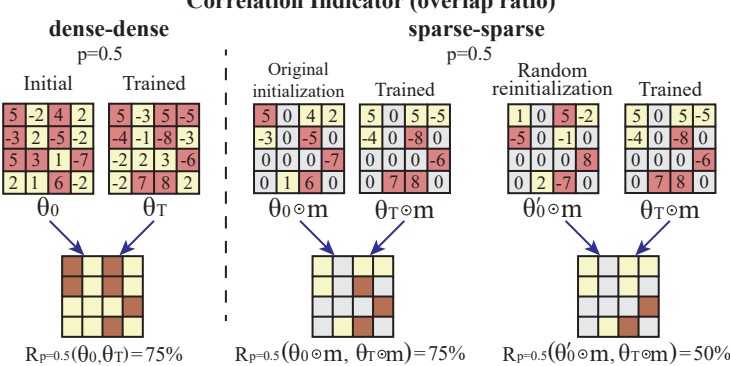

Figure G.1: Scenarios for quantitative analysis of the weight correlation with an example of *sparsity ratio* $= 50\%$ and $p = 0.5$. This example is one DNN layer, while our actual computation is on the whole DNN.

Intuitively, the *weight correlation* means that if a weight is large in magnitude at initialization, it is likely to be large after training. The reason for such correlation is that the learning rate is too low and weight updating is slow. Such weight correlation is not desirable for DNN training and typically results in lower accuracy, as weights in a well-trained DNN should depend more on the location of those weights instead of initialization [9]. Thus when such weight correlation is strong, the DNN accuracy will be lower, i.e., not well-trained.

---

[3]In this way the formula can be unified for dense and sparse DNNs.

[4]We cannot say that there is no correlation here because $R_p(\theta, \theta') \approx p$ is only a necessary condition.