# OpenReview forum: "Sanity Checks for Lottery Tickets: Does Your Winning Ticket Really Win the Jackpot?"
_NeurIPS.cc/2021/Conference — NeurIPS 2021 Poster_

### Official Review · Reviewer_cJrs · 2021-07-10

**Rating:** 8
**Confidence:** 4

**Summary:**

This paper tries to reconcile the controversies around lottery ticket hypothesis. Their study reveals that small learning rate, insufficient training epochs etc. are the main reasons for these controversies. They define a rigorous version of LTH and show that under different training recipes, network architectures and train data size, no rigorous winning tickets are found by the current methods. Their study highlights that using small learning rate on DNNs with residual connections is more likely to find winning ticket and in conditions where small learning rate is not favourable then any special initialisation has no impact on chances of finding a winning ticket.

**Limitations And Societal Impact:**

I think the definition of rigorous LTH can be made more clear. In addition to the empirical explanations, it would be great if these claims can be explained using theory.

**Main Review:**

The paper studies a very important problem and tries to find answers to some of the inconsistencies around LTH. The findings are novel, very insightful and valuable to the LTH and pruning research. Finding a winning ticket is an expensive process and guidance provided by this paper on several hyper-parameters is quite useful. Most part of the paper is well written and has sufficient clarity. I feel the definition of rigorous LTH can be made more clear, possibly using some mathematical constructs. The experiments are done over different datasets( CIFAR-10/100, Tiny-Imagenet) and different architectures (ResNet, VGG, MBNet etc.), which makes their findings more credible. I think the claims provide important hints about LTH however, since the claims are only based on experimental results, so we need to be open to situations where some of their claims may not hold as well.

**Time Spent Reviewing:**

3

---

> ### Author Response · Authors · 2021-08-09
> **Author responses to reviewer cJrs**
>
> We thank the reviewer for the valuable comments and constructive suggestions. Below are our summarized questions and responses.
>
> **Q1: The definition of the rigorous LTH can be more clearer with some mathematical construct.**
>
> **A1**: Thank you for your constructive comments. It is very important to keep the definition clear. We draft a new version of the rigorous definition of the lottery ticket hypothesis with mathematical construct. Please note that the network notations used in the definition are defined in Section 2.1 in our paper.
>
> **The lottery ticket hypothesis -- a rigorous definition.** Suppose that there is a *sub*-network $f(x; m \odot  \theta_0)$ in which
>  the sparse mask $m \in {\\{0,1\\}}^{|\theta|}$ under a non-trivial sparsity ratio
> that is  acquired from
> a certain  pruning algorithm and is associated with  the initial weights $\theta_0$. After $T$-epoch training, let $A_{\mathrm{LT}}$ be the test accuracy achieved by $f(x; m \odot  \theta_0)$.
> Moreover, let $A_{\mathrm{PRE}}$ denote the accuracy of the pretrained dense network from $f(x;\theta_0)$ in a sufficient $T$-epoch training with a decent learning rate.
> Associated with $f(x; m \odot  \theta_0)$,
> let $f(x;\theta^{SD})$ and $f(x;\theta_{0}' \odot m)$ denote a small-dense network with model size the same as $|| m||$ and a randomly reinitialized subnetwork $f(x;\theta_{0}' \odot m)$, with accuracies $A_{\mathrm{SD}}$ and $A_{\mathrm{RR}}$, respectively.
> **The lottery ticket hypothesis is then stated below:**
> $\exists \ f(x;\theta_{0}\odot m)$, when trained with  $T'\leq T$ epochs, can reach to the accuracy   $A_{\mathrm{LT}}$ satisfied with
> $A_{\mathrm{LT}}\approx A_{\mathrm{PRE}} $, $A_{\mathrm{LT}} > A_{\mathrm{SD}}$, $A_{\mathrm{LT}}  >  A_{\mathrm{RR}}$, where $>$ indicates a clear accuracy gap.
>
>
> **Q2: The claims are based on the intensive experiments that give important hints on LTH. Need to be open to the situation that some of the claims may not hold, and it would be great that claims can be explained using theory.**
>
> **A2**: We totally agree with your point. In our revised paper, we will add a section that clearly states the limitations and social impacts of our work and we will add this important statement from your review. The major contribution of our work is to propose a rigorous definition of LTH and principles on how to identify the winning tickets, and our claims are guided by empirical evidence and quantitative analysis. Our intention is to provide a valid experimental basis for subsequent research and to facilitate future work to continue uncovering more advanced experimental methods and theoretical foundations for the lottery ticket hypothesis. We will also continue to work on the theoretical analysis based on the observations in this work.

---

### Official Review · Reviewer_277e · 2021-07-15

**Rating:** 7
**Confidence:** 4

**Summary:**

This paper critically evaluates the lottery ticket hypothesis under different hyper parameter settings. They note that the learning rate used has a substantial impact on the quality of the unpruned model as well as the lottery ticket networks. They propose a stronger form of LTH by requiring the lottery ticket subnetworks to be able to match the best performing unpruned network. They find that using a larger learning rate results in a better unpruned model, but also results in an inability to find equally performant pruned networks.

They show that IMP is better than OMP only for residual networks, and residual networks prefer a smaller learning rate on small datasets, and when larger learning rates are preferred, random reinitialization performs no worse than lottery tickets.

**Limitations And Societal Impact:**

Yes

**Main Review:**

Originality:
- There are some novel observations, including a systematic look at the impact of learning rate on LTH, comparing iterative magnitude pruning (IMP) and one-shot pruning (OMP), as well as the investigation into the loss contours for residual connections. However, the overall originality of the contribution is limited. Some results already exist in prior work. For example:
    - The definition of secondary prize tickets seem very similar to what is used in prior work, and observations regarding them have already been well established in those papers. The failure to train LT from initialization has also been studied in [2].
    - It's well known that initial weights and final weights are correlated, and that LT train to the same basin as the original network, which makes the analysis regarding correlation indicator unsurprising.

Quality:
- The experimental settings in this paper are comprehensive, which I highly appreciate.

- The stricter definition proposed in this paper requires LTs to be found at the best hyper parameter settings for the unpruned network. However, it doesn’t consider the possibility that training the subnetworks would require different optimal hyper parameters. In Lan et al [1], they show that it is optimal to use different learning rates during pre-training (for finding the mask) and during LT training (for training the sparse model), and that doing so leads to well-performing lottery tickets. Though the authors cannot be faulted for not citing non-conference papers, the observations in Lan et al suggest that different learning rates between pruned and unpruned models should also be considered in this work. Intuitively, if a jackpot lottery ticket satisfying the specified conditions can be found by using different learning rates, it should be considered as a successful lottery ticket. Without running those experiments, it raises doubt regarding the soundness of this paper’s main message.

- In section “Which pruning method is better, IMP, OMP , or it does not matter?”, how are the IMP/OMP trained? Is there an unpruned model trained without residual connections, and new lottery ticket masks are derived from that model? If so, how does the loss contours compare for the unpruned models with or without residual connections? I wonder if the smoothing effect is inherent to ResNets, and IMP simply preserves it, or is it specific to sparse models? Do you see the same non-smooth behavior in other non-ResNet model architectures (both unpruned and pruned)? Do these other model architectures prefer larger learning rates (unlike ResNets)?

- No justification is provided for the thresholds of 0.5%, 1%, and 1.5%, other than prior work - please cite the relevant prior work there.

- Missing citations for Stabilizing the Lottery Ticket Hypothesis [2], as many similar experiments have been conducted there. Any differences in results should be noted.

Clarity:
- The paper is overall well written.
- Regarding secondary prize ticket, is it identical to the definition proposed in the original LTH?
- In the section "When does theta_0 benefit subnetwork training?" Could the same conclusions be drawn by simply looking at the correlation of the weights between theta_0 and theta_T?

Significance:
- I think the position of this paper is significant and an important contribution to lottery ticket work. I believe that the stricter form of lottery tickets is necessary in future work and can help remove inconsistencies in this research direction. However, this is contingent on resolving the issues I raised above, especially regarding the hyperparameter optimization performed for lottery ticket networks.

[1] https://openreview.net/forum?id=XkI_ggnfLZ4

[2] https://arxiv.org/pdf/1903.01611v3.pdf

===================================================
Update post rebuttal:

I want to thank the authors for the very thorough response, which helped to address my original concerns. I hope the authors can incorporate many of these discussion points in the final manuscript. On the original reading it seemed like the conclusion was that no jackpot tickets can be found, but the rebuttal provides a more nuanced stance, which I hope can be better reflected in the final paper. I also would encourage the authors to provide a more thorough study of decoupled learning rates (as the authors already indicated that they will do), as it is an important factor in determining final performance and should really be explored in a paper that aims to provide general guidelines for LT training.

Assuming these changes will be incorporated in the final paper, I will raise my score to 7.

**Time Spent Reviewing:**

6

---

> ### Author Response · Authors · 2021-08-09
> **Author responses to reviewer 277e (Part 1)**
>
> **(Part 1 of our response)**
>
> Thank you for your time reviewing our paper. The questions and suggestions are all very constructive. We provide our responses below. Please feel free to let us know if you have additional comments and we are always happy to discuss them here.
>
> **Q1: Secondary prize definition is similar in [2], and [2] also studies the failure of training LT from the initial point.**
>
> **A1**: The definition of Secondary prize ticket (line 176) is similar to the previous LTH work . However, this paper mainly focuses on the quality of the identified ticket (both Jackpot and Secondary) from the **initial point** with different hyper-parameter settings and network structures. We propose a novel definition -- the Jackpot winning ticket to distinguish it from the  definition of “winning” ticket in previous works (Secondary prize ticket). [2] studies the failure of training LT from initial point but [2] mainly discusses why rewinding to an early stage of training can make a successful subnetwork training from the perspective of SGD noise, which is not the same topic as what we have studied in our work.
>
>
>
> **Q2: Correlations exist between initial weights and final weights. The correlation indicator is not surprising.**
>
> **A2**: First of all, we need to mention that the weight correlation defined in our paper is the number of overlapped indices of the top-p · 100% large-magnitude weights between two different sets of weights.
>
> According to our experiments, the weight correlation between initial weight and final weight is conditioned on the learning rate (please refer to line 245, appendix E and Figure 4). When using a relatively small learning rate, the correlation between initial weights and final weights is strong, but both trained networks have relatively low accuracy. In this case, LT is highly likely to train to the same basin as the original network. However, when using a reasonably large learning rate, the correlation is weak, thus LT is likely to train into a basin different from the original network, while the original network usually achieves much higher accuracy. We conjecture that the strong correlation between initial weight and final weight may not be desirable because it means the network weights may not be sufficiently trained, and we think it is a novel point of view.
>
>
> **Q3: Doesn’t consider the possibility that training the subnetworks would require different optimal hyper parameters.**
>
> **A3**: This is a very good question. We are also aware of the possibility of different combinations of learning rates when training the original networks and subnetworks.
>
> **As one major contribution of this work, the rigorous LTH definition and principles for identifying tickets are valid and can hold true for future research.** The reason is that, in our rigorous definition of LTH and principle 5 (line 155), if there exists a learning rate that makes the subnetwork achieve similar or higher accuracy than a well-trained original network, we call the Jackpot winning ticket is identified. However, it is hard to cover all possible settings, so we train the original network and subnetwork using the same learning rate as it is a standard setting in most LTH research, and we explore many different learning rate settings.
>
> Besides different learning rate settings, there is still a large exploration space for future research in this area. We hope our rigorous definition can bring some inspiration to the community and may reconcile existing and potential controversy.
>
> We list some of the preliminary results (ResNet20 on CIFAR10 with sparsity ratio 0.914) of training subnetwork using different learning rates below and we will include more results in our final version. According to our preliminary results, using different learning rates in pretrain and LT train can slightly benefit the accuracy (e.g., 89.7% vs. 89.4% IMP accuracy in the case of pretraining using learning rate 0.01, or 87.4% vs. 87.2% OMP accuracy in the case of pretraining using learning rate 0.1). This further strengthens our claim in line 226, that the Jackpot winning ticket might exist in a network with an appropriate size and trained using a desirable learning rate. We will add your point to our revision and make a clear statement on learning rate setting.
>
>
> **Table R1. ResNet20 on CIFAR10 with sparsity ratio 0.914 at learning rate 0.01**
>
> | Pretrain lr (Acc %) | 0.01        | 90.3        |
> |---------------------|-------------|-------------|
> |        LT lr        | IMP Acc (%) | OMP Acc (%) |
> |        0.001        |     87.5    |     83.3    |
> |        0.005        |     89.7    |     85.3    |
> |         0.01        |     89.4    |     86.5    |
> |         0.05        |     87.9    |     87.2    |
>
>
> **Table R2. ResNet20 on CIFAR10 with sparsity ratio 0.914 at learning rate 0.1**
>
> | Pretrain lr (Acc %) | 0.1         | 92.4        |
> |---------------------|-------------|-------------|
> |        LT lr        | IMP Acc (%) | OMP Acc (%) |
> |         0.01        |     85.3    |     85.8    |
> |         0.05        |     86.6    |     87.4    |
> |         0.1         |     87.3    |     87.2    |
> |         0.15        |     86.7    |     87.3    |
>
>
> **Q4: IMP preserves the smoothness inherited from original residual network?**
>
> **A4**: We construct a residual-free “ResNet-32-like” network which is an original ResNet-32 that removes all residual connections. We pretrain this residual-free “ResNet-32-like” network and derive the mask, then perform IMP or OMP experiments.
>
> Regarding this question, we evaluate the contour of the unpruned model for both original ResNet-32 and the residual-free “ResNet-32-like” network at learning rate 0.01 and 0.1, and we find that ResNet-32 has a smoother landscape, which is consistent with the observation in [18] cited in our paper. We will add these visualizations in our revised paper. According to our observation, it is true that IMP will better preserve smoothness of ResNet-32 than OMP, because the gradual pruning process of IMP may benefit from a smooth contour to effectively explore a smooth route to local minima.
>
> We also notice that under the IMP scenario, a smaller learning rate will further benefit the subnetwork training of ResNet-32. For an unpruned ResNet-32, the landscape basin is larger at learning rate 0.1, which explains the superior accuracy (94.6% at learning rate 0.1 vs. 93.0% at learning rate 0.01). But the subnetwork has better accuracy at learning rate 0.01 with IMP (Figure 5a, 92.9% vs. Figure 5b, 91.4%).
>
>
> **Q5: Non-smooth behavior on other architectures without residual connection? Other architectures prefer larger learning rates (unlike resnet)?**
>
> **A5**: Yes. In our experiments, VGG-16 and MobileNet-v1 do not have residual connections. We evaluate the loss landscape of VGG-16 on CIFAR-10, and a residual network (ResNet-18) that has the similar number of parameters to compare with it. We find that VGG-16 has a more rugged landscape compared to ResNet-18. We will include this comparison in our revised paper.
>
> When the network does not contain residual connections, such as MobileNet-v1 and VGG-16, they all prefer larger learning rates. Please refer to the results in Figure 3, Table 3, and Appendix D for full evaluation data.

---

> > ### Comment · Reviewer_277e · 2021-08-26
> > **Response**
> >
> > I want to thank the authors for the very thorough response, which helped to address my original concerns. I hope the authors can incorporate many of these discussion points in the final manuscript. On the original reading it seemed like the conclusion was that no jackpot tickets can be found, but the rebuttal provides a more nuanced stance, which I hope can be better reflected in the final paper. I also would encourage the authors to provide a more thorough study of decoupled learning rates (as the authors already indicated that they will do), as it is an important factor in determining final performance and should really be explored in a paper that aims to provide general guidelines for LT training.
> >
> > Assuming these changes will be incorporated in the final paper, I will raise my score to 7.

---

> > > ### Author Response · Authors · 2021-08-26
> > > **Author response to reviewer 277e for the follow-up comments**
> > >
> > > We want to thank you for your appreciation of our work and for raising the score!  Your comments are very constructive, e.g., the discussion of the Jackpot winning tickets, the decoupled learning rates for LTH training, and the accuracy drop metric, etc. We will complete the experiments of decoupled learning rates, and we will make sure that all of the discussion points are carefully addressed and included in our final manuscripts, as well as our extra experimental results. The quality of our paper will be improved with your help.
> > >
> > > Thank you again for your valuable time!

---

> ### Author Response · Authors · 2021-08-10
> **Author responses to reviewer 277e (Part 2)**
>
> **(Part 2) Continuing responses.**
>
> **Q6: Accuracy degradation thresholds of 0.5%, 1%, and 1.5% need to be justified by some citation.**
>
> **A6**: Thank you for your suggestion. We have studied many related papers in LTH, DNN pruning, and sparse training. Our accuracy degradation thresholds are mainly summarized from those related works and our experience. For the works that satisfy the thresholds are mostly the state-of-the-art of their research area. We list some of those works here and we will cite them in our revision to make our comparison metric solid. Even though some of the paper doesn’t mention the exact number, we can get an approximate range from the figures or results.
>
> In general, for CIFAR10 dataset, most works achieve accuracy drop less than 0.5% and claim “no accuracy degradation”. For ImageNet-1k, most works have top-1 accuracy drop between 1% to 1.5% and claim “no accuracy degradation”, and we choose 1.5% as a reasonable threshold since ImageNet-1k is a relatively hard dataset. However, it is still hard to find the Jackpot winning ticket on ImageNet-1k dataset.
>
> As our paper proposes a rigorous definition and a guideline to find lottery tickets, we believe an exact metric to measure accuracy loss will be helpful for future research.
>
> *LTH:*
>
> [R1] Zhang, Zhenyu, et al. "Efficient Lottery Ticket Finding: Less Data is More." ICML 2021. (mentioned in the paper: $\leq$0.5% accuracy drop on CIFAR-10)
>
> [R2] Chen, Tianlong, et al. "The lottery tickets hypothesis for supervised and self-supervised pre-training in computer vision models." CVPR 2021. (Refer to the figures in the paper)
>
> [R3]Frankle, Jonathan, et al. "Linear mode connectivity and the lottery ticket hypothesis." ICML 2020. (mentioned in the paper: <0.2%  $\pm$ noise accuracy drop on CIFAR-10)
>
> [R4] Renda, Alex, Jonathan Frankle, and Michael Carbin. "Comparing rewinding and fine-tuning in neural network pruning." ICLR 2020. (already cited. Refer to the figures in the paper)
>
> [R5] Zhou, Hattie, et al. "Deconstructing lottery tickets: Zeros, signs, and the supermask." NeurIPS 2019. (Refer to the figures in the paper)
>
> [R6] Frankle, Jonathan, and Michael Carbin. "The lottery ticket hypothesis: Finding sparse, trainable neural networks." ICLR 2019. (already cited. Refer to the figures in the paper)
>
> *Sparse training:*
>
> [R7] Evci, Utku, et al. "Rigging the lottery: Making all tickets winners." ICML 2020. (refer to the figures)
>
> [R8] Jayakumar, Siddhant M., et al. "Top-kast: Top-k always sparse training."NeurIPS 2020 (refer to figures).
>
> [R9] Mostafa, Hesham, and Xin Wang. "Parameter efficient training of deep convolutional neural networks by dynamic sparse reparameterization." ICML 2019. (refer to table and figures)
>
> *DNN pruning:*
>
> [R10] Lin, Tao, et al. "Dynamic model pruning with feedback." ICLR 2020. (refer to the results)
>
> [R11] Gale, Trevor, Erich Elsen, and Sara Hooker. "The state of sparsity in deep neural networks." arXiv preprint arXiv:1902.09574 (2019). (refer to the results)
>
> [R12] Zhu, Michael, and Suyog Gupta. "To prune, or not to prune: exploring the efficacy of pruning for model compression." ICLR workshop 2017. (mentioned in the paper: 2% top-5 accuracy drop on ImageNet)
>
>
> **Q7: Missing citation of [2]. Need to discuss the difference.**
>
> **A7**: Thank you for pointing it out. We will cite [2] in our revision. The citation [2] studies a subnetwork that is stable to SGD noise can train to the same accuracy as a dense network. It makes a key observation that subnetworks are stable to SGD noise in early stage of training, which explains why rewinding technique succeeds in LTH. In our paper, we recognize rewinding technique as a successful approach (please refer to appendix A and B) to match accuracy, but we distinguish this method with training subnetwork from initial point because they are two different topics (line 179). We focus on **initial point** because it is more fundamental to the network initialization state and topology, and we propose a rigorous LTH definition and corresponding principles to identify two different types of winning tickets. We focus on the effects and their rationales of hyper-parameter settings, network structures, pruning methods and the size of the training dataset, which are all distinctive to [2]. Our definition, results, quantitative analysis and guideline suggest that Jackpot winning tickets are highly likely to exist, and will promote new methods other than IMP, OMP, or rewinding techniques in future research.
>
>
>
> **Q8: Could the same conclusions be drawn by simply looking at the correlation of the weights between $\theta_{0}$ and $\theta_{T}$?**
>
> **A8**: Yes. When the correlation between $\theta_{0}$ and $\theta_{T}$ is positive, the accuracy of subnetwork and pretrain dense network is more close. However, please note that the positive correlation may not be desired for subnetwork training because it means the learning rate could be unreasonably small and the weights may not be sufficiently trained.

---

### Official Review · Reviewer_Dhgf · 2021-07-16

**Rating:** 7
**Confidence:** 4

**Summary:**

This paper goes deep into the investigation of the lottery ticket hypothesis, and found out that the training recipe that the original findings were based on had been insufficient. With a revised recipe and through extensive experiments, the author empirically studied the patterns of whether and when a "jackpot ticket" and "secondary tickets" can be identified. The major factor turns out to be the learning rate with which to train the subnetworks. The paper then ends with guidelines on how to find winning tickets.

**Limitations And Societal Impact:**

The author claims that they found a better training recipe than used in original LTH. And Figure 1 is the only figure that showed such superiority. However it is on three hand-picked sparsity values. One plot I feel like is missing and could really drive this point home is one that plots accuracy vs. sparsity like in the original LTH and many follow-up works. So instead of picking three sparsity levels, the three subplots would be the three LRs, and each subplot would have x axis being the sparsity level or "percentage of weight remaining" like in all the other LTH papers.

Another limitation that invites questions is the ImageNet result. According to paper, "Tiny-ImageNet and ImageNet-1K, no clear secondary prize tickets are identified using ResNet-18 or ResNet-50." That is not quite believable. One possibility is that the author adopted the original rewinding recipe used for CIFAR-10, where the weights are rewound to initial values. However a follow-up work on LTH on larger models [1] demonstrated that it is necessary to rewind to a later point during training for IMP to succeed. I suggest the author study this paper and many other papers on finding winning tickets large-scale models and datasets, which almost all proved the existence of winning tickets in ImageNet and ResNets.

[1] Jonathan Frankle, Gintare Karolina Dziugaite, Daniel M Roy, and Michael Carbin. Stabilizing the
lottery ticket hypothesis. arXiv preprint arXiv:1903.01611, 2019.

Errors, modifications or improvements needed:
 - In 2.1 when defining a network function f (·), I don't think input x should be included, that is, it should just be f (\theta) as opposed to f(x, \theta). Both because the function shouldn't depend on input, but also throughout the section x is always the same and hence repetitive.
 - Figure 2 the difference between (b) and (c) is unclear. According to the definition, the highlighted difference between "jackpot" and "secondary" should be whether the original network is well trained (with appropriate LR and length of training) or not. The plots though seem to only highlight whether the crossing point is at a lower or higher LR. Therefore it doesn't seem consistent with the definition.

Minor:
Each figure and table should be self-sufficient, in that the caption should convey what the reader is supposed to get out of. Sometimes that requires repeating what's said in the main text but it should be very much encouraged. For example Figure 6 is confusing to me as what observation or conclusion we are supposed to draw.



**Main Review:**

The paper challenges the established LTH, reproduced and improved the original training recipe, and comes up with a revised definition, and recommended guidelines on obtaining winning tickets based on an extensive set of empirical study. Overall I think such a sanity check endeavor is of great value to the community.


**Time Spent Reviewing:**

3

---

> ### Author Response · Authors · 2021-08-10
> **Author responses to reviewer Dhgf**
>
> We would like to thank the reviewer's efforts in reviewing our paper and providing many important comments. We summarize the questions in the review and propose our responses below.
>
> **Q1: Do you have accuracy vs. sparsity curve, like the ones in the original LTH paper?**
>
> **A1**: This is a very good suggestion. We do have all the accuracy data for every sparsity level starting from overall 20% to 95.6% (i.e. 20%, 36%, 48.8%, 59%, 67.2%, 73.8%, 79%, 83.2%, 86.6%, 89.3%, 91.4%, 93.1%, 94.5%, 95.6%) because IMP needs to traverse all of them by pruning 20% of the unpruned weights at each iteration. We also evaluate 6 different learning rates for all those sparsity levels with different LTH settings (line 77). The reason that we pick 3 sparsity levels is that this paper mainly focuses on the influence of the training setting of LTH and there’s limited space of the main body of the paper. We want to make the comparison clearer so we only sample 3 representative sparsity levels (low - 59%, medium - 83.2%, high - 91.4%), while all other sparsity have consistent observations. We will include the accuracy vs. sparsity curve in the main body and the appendix in our final version.
>
>
> **Q2: Secondary prize tickets on Tiny-ImageNet and ImageNet using RN18 and RN50 can be found using rewinding to a later point [1], but the author claims it can not be identified, why?**
>
> **A2**: Thank you for your suggestion. We actually have the discussion and the corresponding results regarding the rewinding settings. We also generalize our rigorous definition of LTH to rewinding ticket settings. Please refer to line 179-184 and Appendix A & B.
>
> As we mentioned in the paper (line 179), we distinguish our definition of the lottery ticket hypothesis from the weight rewinding technique. The initial point subnetwork training is a study of the combination of initialization and network topology, while rewinding to an early point is an approach that quickly and efficiently restore subnetwork accuracy.
>
> We believe that the initial point of a network is unique (i.e., unlike rewinding to an arbitrary training stage), and the subnetwork associated with it has a certain significance on sparse network training, especially for training at resource-limited scenes, because it doesn’t involve training a dense network before applying sparsity.
> Under our rigorous definition of LTH, secondary prize tickets are not identified on a large-scale dataset with RN18 and RN50 since we excluded rewinding technique based on the above considerations.
>
> Current LTH research is mainly based on IMP method to find sparse mask, and we believe our rigorous definition of LTH and guideline in this paper can potentially promote further research to build upon it, for *another approach (parallel to rewinding)* to find sparse mask and successfully train a subnetwork from the initial point.
> Under our generalized definition for rewinding tickets in Appendix A, the rewinding ticket can be identified, as shown in Appendix B. So our results are not contradictory to the prior works.
>
>
> **Minor issue 1: $f(\theta)$ instead of $f(x; \theta)$ in Sec 2.1**
>
> **A3**: Thank you for pointing it out. We intended to be consistent with the original LTH paper ([1] in our citation). We will remove the “x” in our final revision.
>
>
> **Minor issue 2: Difference between (b) and (c) in Figure 2 is unclear.**
>
> **A4**:  This is a very constructive suggestion. We will modify Figure 2 (b) and (c) to be clearer.
>
>
> **Minor issue 3: Each figure and Table should have a detailed description.**
>
> **A5**: Thank you for this is a good suggestion. We will add detailed descriptions for each figure and table in our final revision.

---

### Decision · Program_Chairs · 2021-09-27

**Decision:**

Accept (Poster)

**Comment:**

This paper investigates the impact of hyperparameters on lottery ticket performance and uses these analyses to provide guidance as to how to best select these hyperparameters. All 3 reviewers praised the paper for its comprehensive and compelling experiments as well as its clarity. There was some concern regarding the stringent definitions used in this work, but clarifications in the discussion were sufficient to resolve these concerns so long as the authors add additional discussion to the manuscript. Given the rigor of this work and its relevance to the lottery ticket community, I recommend this paper is accepted.